# Orthogonal inducible control of Cas13 circuits enables programmable RNA regulation in mammalian cells

Yage Ding[1], Cristina Tous[1], Jaehoon Choi[1], Jingyao Chen [1] & Wilson W. Wong [1] ✉

RNA plays an indispensable role in mammalian cell functions. Cas13, a class of RNA-guided ribonuclease, is a flexible tool for modifying and regulating coding and non-coding RNAs, with enormous potential for creating new cell functions. However, the lack of control over Cas13 activity has limited its cell engineering capability. Here, we present the CRISTAL (Control of RNA with Inducible SpliT CAs13 Orthologs and Exogenous Ligands) platform. CRISTAL is powered by a collection (10 total) of orthogonal split inducible Cas13 effectors that can be turned ON or OFF via small molecules in multiple cell types, providing precise temporal control. Also, we engineer Cas13 logic circuits that can respond to endogenous signaling and exogenous small molecule inputs. Furthermore, the orthogonality, low leakiness, and high dynamic range of our inducible Cas13d and Cas13b enable the design and construction of a robust incoherent feedforward loop, leading to near-perfect and tunable adaptation response. Finally, using our inducible Cas13 effectors, we achieve simultaneous multiplexed control of multiple genes in vitro and in mice. Together, our CRISTAL design represents a powerful platform for precisely regulating RNA dynamics to advance cell engineering and elucidate RNA biology.

RNA plays a vital role in diverse critical cellular processes, such as disease development and cell differentiation[1,2]. Precise and dynamic control of RNAs is increasingly needed to understand their biology. For example, inducible up- or down-regulation of long non-coding RNAs (lncRNA) is often used to interrogate their function in diverse cell types[3–5]. Furthermore, it would be powerful to simultaneously and orthogonally regulate multiple RNAs to unravel their complex interactions and engineer sophisticated gene circuits[6–8], which could lead to breakthroughs in a range of biotechnological and medical applications[8–10].

Recently, DNA level control of transcription, such as through synthetic transcription factors or genome editing tools, has advanced substantially[11–13]. While useful, these tools can be limited by their specificity targeting RNA splicing isoforms, their targeting ability that is affected by chromatin regulations or sequence restriction, and their safety due to risk of permanent aberrant DNA modifications[14–16]. As

such, alternative tools are needed to exert safer and more precise control over RNAs.

Regulatory tools that target RNA, as opposed to DNA, are safer and have a broader target sequence space, making them an attractive cell engineering tool and therapeutic modality[17–19]. Current established RNA level regulatory tools can be broadly classified as cis- or trans-acting[20]. Cis-acting elements, such as riboswitches or protein binding motifs (e.g., L7Ae), are RNA motifs embedded within transcripts that regulate RNA translation or degradation in a ligand-responsive manner[14,21]. Although these motifs are versatile in generating higher-order networks, they require genome modification to gain control over the endogenous transcriptome. In contrast, trans-acting RNA switches that utilize the RNA interference (RNAi) pathway allow control of endogenous transcript expression without genome modification[22,23]. However, only a few ligand-responsive modules can synergize with RNAi, and they tend to have a small dynamic range[24–27].

[1]Department of Biomedical Engineering, Biological Design Center, Boston University, Boston, MA 2215, USA. ✉e-mail: wilwong@bu.edu

To address this tradeoff between endogenous transcript targetability and versatile ligand-responsive control, we took advantage of the CRISPR/Cas13 system[28–31].

Cas13 is a class II type VI CRISPR effector that relies on guide RNA (gRNA) complementarity to target an RNA transcript for cleavage[28–31]. Cas13 can potentially target any sequence, similar to RNAi-based tools. Also, the enhanced specificity of Cas13 and the nuclear localization capability enable it to target other types of RNAs, such as viral RNAs[32], circular RNAs[23] and nuclear RNA (e.g., long-noncoding RNAs lncRNAs)[28]. Additionally, coupling Cas13 with other protein modules such as RNA modification domains or fluorescent proteins allows RNA editing and dynamic imaging of RNA in living cells[28,29,31,33,34]. Recognizing the great potential of Cas13, we developed a platform that we called CRISTAL (Control of RNA with Inducible SpliT CAs13 Orthologs and Exogenous Ligands), which consists of a collection of Cas13-based RNA regulatory circuits for dynamic and orchestrated transcriptome engineering.

To control Cas13 activity, we systematically identify split sites within multiple Cas13 proteins that can be inducibly dimerized to reconstitute their activity (Fig. 1A). This inducible dimerization approach is advantageous over other means of regulating effector function because it is fast-acting, tunable, reversible, scalable, cell type- or state-independent, and composable with other modes of regulation, such as transcriptional and post-translational control[35,36]. Using several heterodimerization systems[28,29,31,37–39] and Cas13 orthologs, we create 10 different chemical-inducible systems that offer efficient knockdown at the protein level up to 78% in vitro and up to 98% in mice against an abundantly expressed transgene using nontoxic plant hormones or FDA-approved drugs. To showcase the versatility of our Cas13 platform, we implement AND logic that only cleaves RNA transcripts in the presence of both an exogenous small molecule and an endogenous pathway-activating signal. We also apply our systems for multiplex and demultiplex regulation and implement an incoherent feed-forward loop utilizing two orthogonal inducible Cas13 effector to create a near-perfect and tunable adaptation response. As the output can be easily switched from a transgene to an endogenous gene, our system can render the endogenous transcriptome under complex regulation. Finally, we achieve orthogonal regulated multiplex RNA knockdown in mice, further demonstrating the versatility and potential of our system (Fig. 1B).

## Results

### Screening RfxCas13d split sites for inducible activity using gibberellic acid inducible dimerization domains

We chose to engineer Cas13d derived from *Ruminococcus flavefaciens* XPD3002 (RfxCas13d) due to its compact size and robust activity in mammalian cells[28]. To predict the structure of RfxCas13d, we aligned its amino acid sequence to its close homolog, the Cas13d from *Eubacterium siraeum* (EsCas13d) whose protein structure has been fully resolved[40]. Based on the prediction, 27 split sites were chosen to avoid secondary structures and conserved regions (Fig. 1D). We used the gibberellic acid (GA)-inducible dimerization domains (GID and GAI) to screen for the inducible activity of the selected split sites since they have demonstrated robust activation for many effectors including recombinases and transcription factors[36,38]. We designed our GA-inducible split proteins with the N-terminal moiety of split Cas13d linked to the GID domain by a GS linker, and the C-terminal moiety led by the complementary GAI domain also using a GS linker for connection (Fig. 1A, Supplementary Fig. 1A). Since RfxCas13d is most efficient when targeted to the nucleus in mammalian cells, we targeted both moieties to the nucleus using the SV40 nuclear-localization signals (NLSs) (Fig. 1C, Supplementary Fig. 1A).

To test the activity of our split Cas13d collection, we transfected HEK293FT cells with plasmids encoding the GA-inducible split Cas13d pairs, an mCherry (mCh) targeting gRNA, mCh, and iRFP as a transfection marker. We measured the mCh and iRFP mean fluorescent intensity (MFI) via flow cytometry (Fig. 1C). Two attributes of a potent inducible system are strong induction and low leakiness (Supplementary Fig. 1B, C). We defined system leakiness as the difference between 1, and the relative mCh MFI of the uninduced state, representing mCh specific knockdown caused by the split Cas13d in the absence of GA (Supplementary Fig. 1B). Inducibility is defined as the difference between the relative mCh MFI of an induced and uninduced condition, normalized to the uninduced condition (Supplementary Fig. 1B). Therefore, an ideal inducible system should have leakiness = 0 and inducibility = 1 (100% knockdown) (Supplementary Fig. 1B, C).

The initial screening identified a few inducible systems with high leakiness (Fig. 1D). Out of 27 screened split sites, 10 generated inducibility larger than 0.4 (Fig. 1D, E). Moreover, inducibilities greater than 0.6 were associated with high leakiness (Fig. 1D, E). While the top inducibility was 0.74 with a leakiness of 0.23 offered by split 263/264, many other split sites with inducibility higher than 0.6 suffered from leakiness up to 0.6, (e.g., split 88/89) (Fig. 1D, E). This split's highest constitutive activity is possibly due to its location at the end of the N-terminal domain (NTD)[40]. This domain clamps with the C-terminal HEPN2 domain on the direct repeat region of the Cas13d gRNA, which may result in a gRNA-induced dimerization of the N-terminus split domain without the inducer reconstituting Cas13d[40]. Without recruitment domains, split sites with great leakiness in this first screen spontaneously reconstituted and became highly active in the presence of mCh targeting gRNA, while none of the fragments have endonuclease activity alone (Supplementary Fig. 2). Many inducible split sites, such as 177/178, 180/181, and ones from 507/508 to 583/584, are predicted to be in regions on the surface of the enzyme whose omission did not reduce the catalytic activity of the WT EsCas13d[40]. Therefore, inhibition of the auto-assembly of the split Cas13 is needed to reduce leakiness.

### Optimization for high-performance systems

In addition to split sites, the location of the CIDs and the localization signal can also impact the switch performance (Fig. 1A, E, F). We hypothesized that orientations of the inducible heterodimerization domains relative to the Cas13d splits might result in steric hindrances that can potentially block spontaneous dimerization (Fig. 1E). Thus, we inspected split sites with inducibility greater than 0.4 in 2 orientations where domain GID and GAI are connected to either the N- or C-terminal of split Cas13 moieties (Fig. 1E, Supplementary Table 1). Interestingly, the reversed orientation, N-GAI/GID-C, with split 507/508 achieved robust inducibility of 0.82, comparable to the WT RfxCas13d knockdown efficiency, with minimal leakiness (Fig. 1E, Supplementary Fig. 3, Supplementary Table 1). Moreover, this orientation generated substantial improvement in system performances with many split sites than the original (Supplementary Table 1). Thus, by reversing the orientation of the CIDs, we not only generated an ideal GA-inducible split Cas13d, but also identified the potentially preferred orientation in designing future split Cas13 ribonucleases.

Besides this approach, we also hypothesized that the spatial sequestration of the split moieties in different cellular compartments would prevent spontaneous dimerization, thus reducing system leakiness[35] (Supplementary Fig. 4A). We attached HIV nuclear export signals (NESs) to target one split moiety to the cytosol while leaving the other moiety sequestered in the nucleus as in the original design (Fig. 1F, Supplementary Fig. 4A). Of the different designs we tested, the N moiety tagged with 2 NESs and the C moiety tagged by 2 NLSs rescued leakiness of different levels in multiple designs (Fig. 1F, Supplementary Fig. 4B–D). Through screening and optimization, we designed multiple GA-inducible split RfxCas13d systems with minimal leakiness and an inducible knockdown efficiency approaching WT Cas13d.

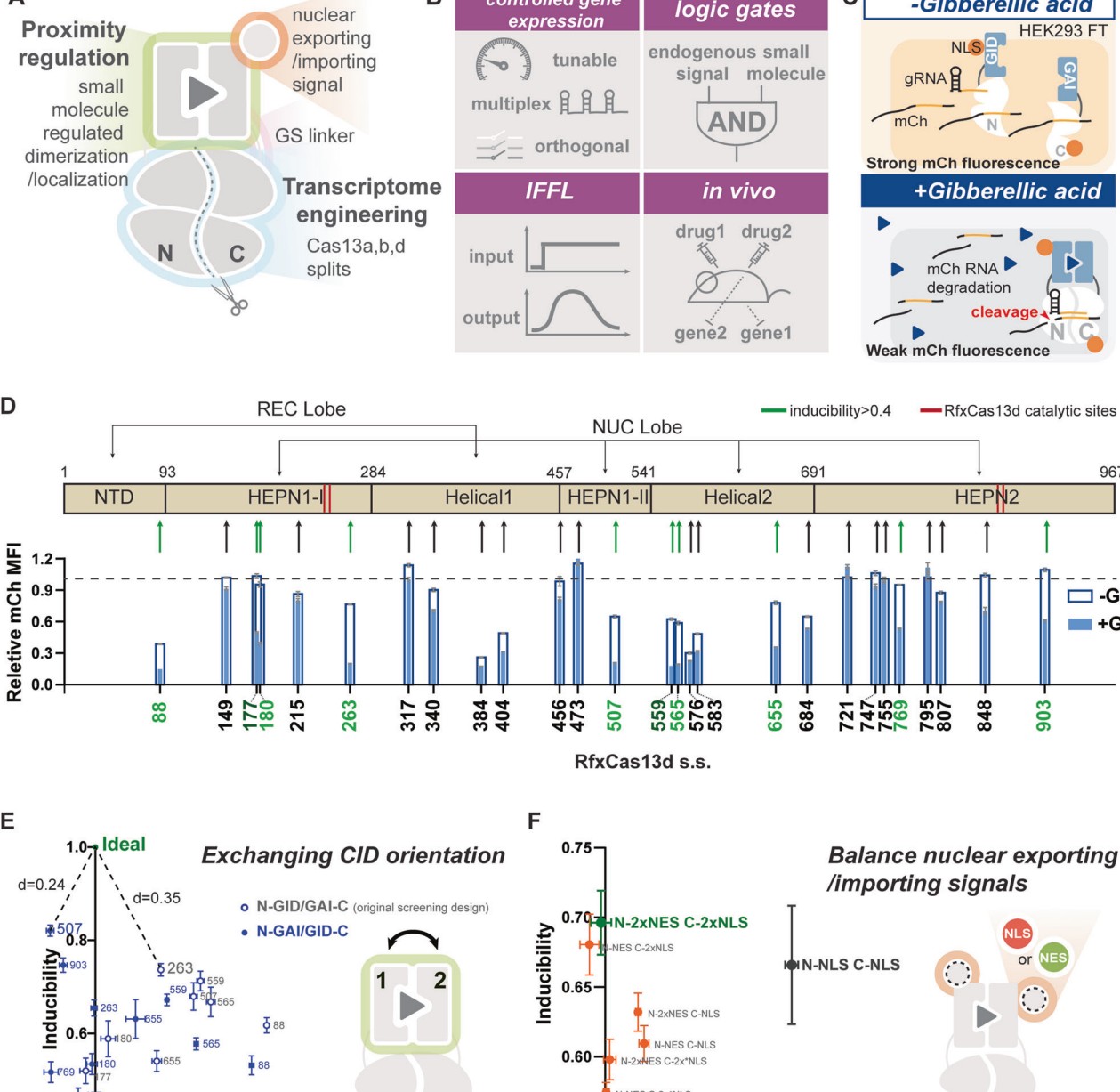

**Fig. 1 | Designing inducible split Cas13 for orthogonal, multiplex, and composable RNA knockdown. A** The small molecule-inducible split Cas13 design comprises a small molecule-inducible dimerization/localization system, cellular compartment localization signals (NLS/NES), GS linkers, and split Cas13 ribonucleases. Reconstitution of Cas13 fragments is regulated by the small-molecule-induced dimerization/localization system in response to corresponding ligands. **B** Distinct features of the split Cas13 ribonuclease collection. The designed systems are tunable and achieve orthogonally regulated multiple transcript knockdown. When integrated with transcriptional control responsive to an endogenous pathway, the design system forms AND logic regulated by both endogenous and exogenous small molecules. When cascaded with other systems in the collection, they formed an incoherent feedforward loop (IFFL) where output expression adapts in response to a sustained small molecule signal. Finally, the designed system displayed robust orthogonally regulated multiplex knockdown in vivo. **C** The schematic shows the screening experimental setup. HEK293FT cells were transfected with plasmids encoding the GA-inducible split Cas13d systems, a targeting gRNA, mCh, and iRFP. Without the inducer, GA, transfected cells should express mCh strongly. When induced with GA (blue triangle), Cas13d splits should be

reconstituted and become active to cleave mCh RNA resulting in weak mCh fluorescence. **D** Split sites were selected throughout the sequence of RfxCas13d in all domains. Ten out of 27 split sites screened with the GA-inducible dimerization domains generated inducibility >0.4 and suffered from leaky activity under no GA conditions. Data are presented as mean values +/− SEM. **E** Orientation of the GA-inducible dimerization domains was exchanged for GA-inducible split RfxCas13d designs with >0.4 inducibility. The orientation N Cas13d-GAI/GID-Cas13d C yielded higher inducibility with more split sites tested than the other orientation. And the design N507 Cas13d-GAI/GID-Cas13d 508 C had good performance with >0.8 inducibility and 0 leakiness. Data are presented as mean values +/− SEM. **F** Tagging N559 Cas13d with 2 NESs and 560 C Cas13d with 2 NLSs completely eliminated leaky activity generated with split N559/560 C in the original screening design while generating no negative impact on the inducibility. Tagging N559 Cas13d with an NES on the C-terminal achieved a similar performance, while using other combinations of NES and NLS led to reduced inducibility. The schematic shows the possible locations of NES or NLS for each design. Data are presented as mean values +/- SEM. Error bars indicate the SEM for three biological replicates ($n = 3$). Source data are provided as a Source Data file.

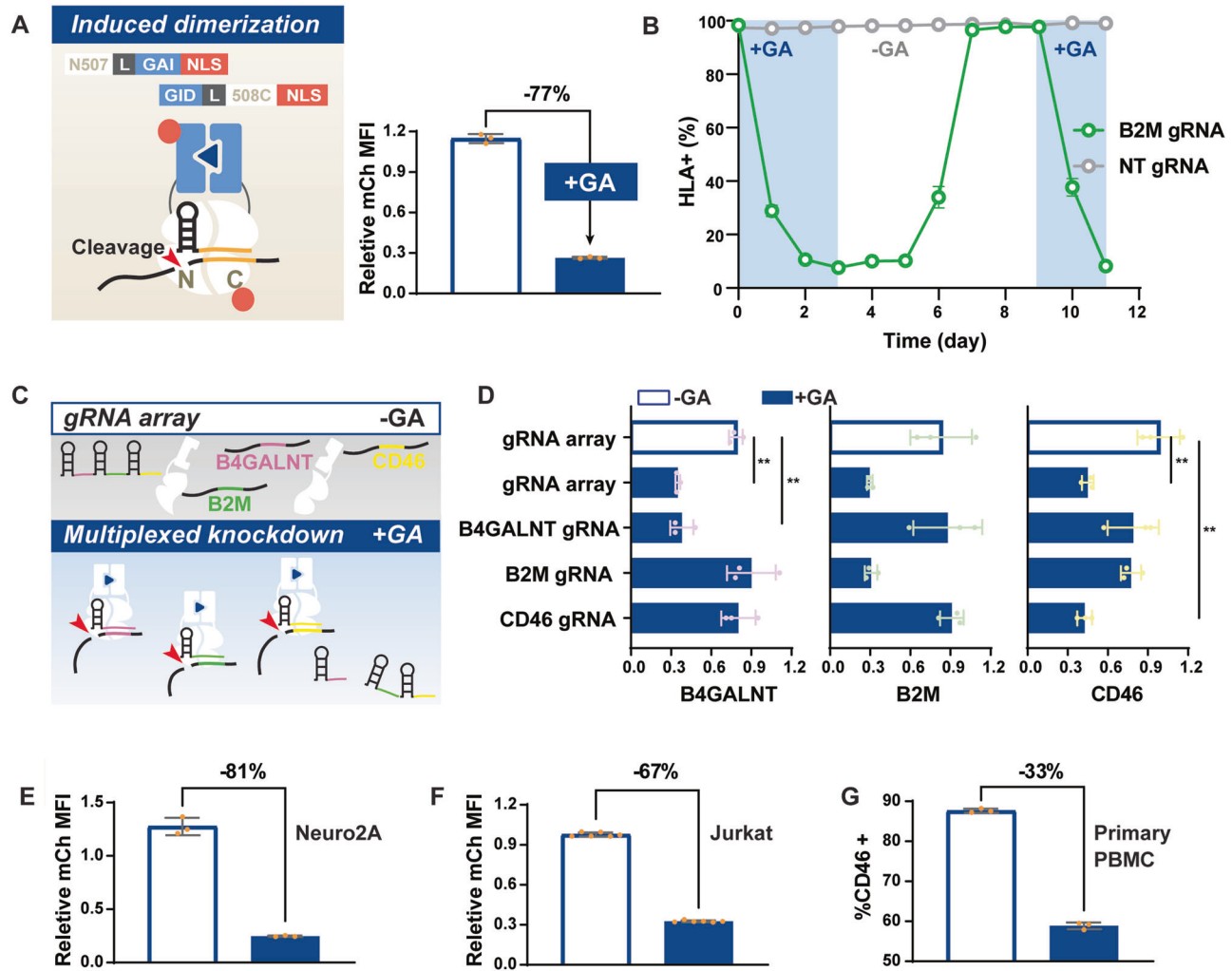

**Fig. 2 | Characterization of the GA-inducible split Cas13d (N507/508 C). A** The schematic shows the detailed design and reconstituted state of N507-GAI-NLS/GID-508C-NLS in the presence of GA, gRNA, and target transcripts. This design achieved 77% mCh-specific knockdown in transient transfection conditions. **B** GA-induced endogenous gene knockdown N507-GAI-NLS/GID-508C-NLS is reversible. HEK293FT cells were integrated with a PiggyBac transposon expression cassette containing the inducible Cas13. The sorted clones that highly express the inducible system were transduced with non-target (NT) gRNA or B2M targeting gRNA lentivirus. B2M knockdown was measured as HLA surface expression. The percent of HLA-positive cells were gated using an unstained control on the day of flow cytometry. The entire population lost HLA expression in 2–3 days of GA induction, which was fully recovered by day 4 after the withdrawal of the inducer. A second GA induction robustly led to the loss of HLA expression. Sample size $n = 3$ biological replicates. **C** A schematic showing the gRNA array processing by split Cas13. Guide RNA arrays were expressed as a single transcript containing spacers joined together with RfxCas13d-specific direct repeat regions. Under GA-induction, split Cas13d becomes reconstituted. They are then actively coupled with processing gRNA arrays to mature gRNAs for targeted knockdown of specific transcripts. Sample size $n = 3$ biological replicates. **D** With a gRNA array targeting B4GALNT, B2M, and CD46, the GA-inducible split Cas13d knocked down all 3 target expressions in a GA-responsive manner with efficiency comparable to that achieved using the single mature gRNA. In addition, the GA-induced single target knockdown was specific as the expression of untargeted transcripts is intact; *P*-value from left to right: $p = 0.00157$, $p = 0.00226$, $p = 0.00507$, $p = 0.00467$. **E** GA induced over 80% mCh knockdown in a neuron cell line, Neuro2A, in transient transfection conditions. Sample size $n = 3$ biological replicates. **F** GA induced 67% mCh knockdown in an immune cell line, Jurkat T cell, in transient transfection conditions. Sample size $n = 6$ biological replicates. **G** Primary cells, PBMCs, were sequentially transduced with 2 lentiviruses, each carrying the N-terminal construct or the C-terminal construct with a gRNA expression cassette. CD46 surface expression was inducibly knocked down in 33% of transduced PBMC. Sample size $n = 3$ biological replicates. *P*-values were calculated as a 2-tailed *t*-test. Data are presented as mean values +/− SEM. Error bars indicate the SEM for at least three biological replicates ($n \geq 3$). Source data are provided as a Source Data file.

## Characterization of GA-inducible split Cas13d

A key feature of an inducible system is tunable regulation in activity and time (Fig. 2A, B, Supplementary Fig. 5G, Supplementary Fig. 16D). Our dose-response experiment revealed fully tunable mCh knockdown efficiency spanning ~4 logs of GA concentrations (Supplementary Fig. 16D). GA-inducible split Cas13d is highly sensitive and can reach full activity at sub-micromolar concentration with an EC50 of 1.9 nM (Supplementary Fig. 16D). Our inducible system is also completely reversible, which is beneficial for in vivo applications such as a safety switch. To study the reversibility of the system, we monitored the knockdown of an endogenous gene beta-2 microglobulin (B2M), a

critical component of the human leukocyte antigens (HLA), over time. We first verified the GA-induced knockdown of B2M in transient transfection settings (Supplementary Fig. 5A). The inducible Cas13d can down-regulate HLA surface expression in 50% of transfected cells in response to GA, similar to the knockdown achieved by WT Cas13d (Supplementary Fig. 5A). For the reversibility experiment, we stably integrated the inducible Cas13d in HEK cells using the PiggyBac transposon system, followed by antibiotic selection, and cell sorting for highly Cas13d-expressing cells (Supplementary Fig. 5C). We observed over 80% GA-induced B2M transcript knockdown when we transfected these cells with B2M targeting gRNA (Supplementary

Fig. 5B). To achieve stable expression of gRNAs, we transduced the effector expressing cell line with lentivirus carrying a B2M targeting or a non-targeting gRNA expressing cassette (Supplementary Fig. 5C). After verification of stable expression of the system components, HLA expression was monitored during a cycle of GA induction and withdrawal (Supplementary Fig. 5C). Results showed that the entire population lost HLA expression within 2 to 3 days of GA induction (Fig. 2B, Supplementary Fig. 5G). More importantly, this knockdown recovered 100% back to the pre-induction level at the protein level 4 days after removing GA (Fig. 2B, Supplementary Fig. 5G). HLA expression was specifically targeted for knockdown during induction. Expression of bystander fluorescence markers we used to monitor Cas13 split expression (BFP and iRFP) and gRNA integration (GFP) were not affected and expressed consistently (Supplementary Fig. 5D–F). Furthermore, the system's activity can be induced a second time to achieve the same dynamic range and efficiency (Fig. 2B).

Beyond the above benefits of tunability and reversibility, Cas13 can intrinsically process gRNA arrays, which allows multiplexed knockdown (Fig. 2C). This property would be useful for the dynamic regulation of a network of genes by our inducible Cas13d. However, the gRNA array processing ability resides in a region distinct from the RNA cleavage in the Cas13 protein. Whether the split Cas13 enzymes preserve their guide-array processing ability is uncertain[40]. Therefore, we validated this function by constructing a gRNA array targeting B4GALNT, B2M, and CD46 (Fig. 2C). The induced split Cas13d achieved knockdown of all 3 targets with the gRNA array as efficient (50–70% efficiency) as with the single target gRNAs (Fig. 2D, Supplementary Fig. 6). The induced knockdown of each target did not affect the expression of the other 2 untargeted genes (Fig. 2D, Supplementary Fig. 6). Additionally, the comparable level of CD46 knockdown by its single targeting gRNA and the gRNA array where the CD46 spacer is located in the third place implies the preservation of efficient gRNA array processing activity by the split Cas13d 507/508 (Fig. 2D, Supplementary Fig. 6).

To demonstrate the portability among different mammalian cell types, we introduced our inducible Cas13d into mouse Neuro-2a, Jurkat T, and human primary T cells (Fig. 2E–G). The inducible system successfully maintained its high performance when transiently transfected in cell lines (Fig. 2E, F). The performance in primary T cells at targeting endogenous gene expression on the protein level was reduced, possibly due to limited expression level and insufficient delivery of system components (Fig. 2G).

## Multiplex regulation of split Cas13d

Beyond the GA-inducible split Cas13d, split Cas13d systems regulatable by orthogonal ligands would further increase their flexibility. As such, we developed four orthogonal inducible systems (three inducible activations, and one inducible inhibition) using abscisic acid (ABA) inducible dimerization domains ABI and PYL[37], tamoxifen (4-OHT) inducible nuclear localizing ER$^{T2}$ domain, Danoprevir (Dano) inducible dimerization domains dNS3 and DNCR[39], and Dano-inhibited dimerization domains dNS3 and ANR[39] (Fig. 3A, Supplementary Fig. 7, Supplementary Table 2). The Dano-inhibited dimerization domains offer an opportunity to turn off Cas13 activity in the presence of Dano.

We focused our designs using only split sites that yielded GA-inducible systems. Split sites that have considerable GA inducibility also show highly efficient inducible knockdown using other inducible systems (Supplementary Fig. 7). This suggests that strong-performing split sites can be prioritized in screening for future designs with more inducible systems, reducing the effort of this process.

While we observed inducibility across different CID systems, leakiness also occurred for other systems, confirming our previous finding that spontaneous reconstitution of the Cas13 splits is not caused by the recruitment domains (Supplementary Figs. 2, 7). However, changing the CID orientation did not rescue the leakiness activity of split

Cas13d equipped with the 4 distinct inducible domains (Supplementary Fig. 8). Although the best performance generated through initial screening offered great inducibility, higher than 0.7 in ABA and Dano-regulated systems, they suffered from substantial leakiness greater than 0.3 (Supplementary Fig. 8A, C, D). In both Dano-regulated systems, inducibility was reduced across all split sites when the C termini of dNS3 were fused with Cas13d C-terminal fragments (Supplementary Fig. 8C, D). This is likely due to the fusion of Cas13 fragments disturbing the structured C-terminal alpha helix of dNS3 in both Dano-regulated systems[39]. In contrast, both termini of the GA and ABA CIDs are unstructured and relatively open for the fusion of Cas13 fragments[41,42]. Therefore, both orientations for these CIDs generated inducible split Cas13d systems with inducibility greater than 0.4 (Fig. 1E, Supplementary Table 1, Supplementary Fig. 8A). However, we still observed that the orientations N-GAI/GID-C, N-ABI/PYL-C, and N/ER$^{T2}$-C generated more ideal performances across more Cas13d splits than their reversed orientations, suggesting CID terminal structure is not the only contributing factor for strong inducibility (Fig. 1E, Supplementary Table 1, Supplementary Fig. 8A, B). Additionally, split 507/508 consistently generated the best performance with the GA-inducible, ABA-inducible, Dano-inducible, and Dano-inhibited systems (Figs. 1E, 3A, Supplementary Table 1, Supplementary Fig. 8A, C, D).

While valuable design rules were revealed, system performance was still not ideal. Therefore, we further optimized the top design of all inducible systems by tuning their localization in cellular compartments (Supplementary Fig. 9). For ABA and Dano-inhibited systems, tagging N-terminal moieties with one NES and C-terminal moieties with 2 NLSs reduced leakiness without reducing inducibility (Supplementary Fig. 9A, B, G, H). Since the ER$^{T2}$ domain for the 4OHT-inducible system is naturally retained in the cytosol in the absence of 4OHT, we did not tag this design with extra NESs. Instead, we explored the addition of another ER$^{T2}$ domain and the possible locations of the NLS. We identified a 4OHT-inducible system with 0.6 inducibility and minimal leakiness (Supplementary Fig. 9C). Meanwhile, since split 507/508 consistently offered strong performance with different CIDs (Fig. 3A, Supplementary Fig. 8A, C, D), we also optimized the design N507/ER$^{T2}$–508C-NLS, despite its inferior performance in original screening (Supplementary Figs. 8B, 9C). As expected, optimized 4OHT-inducible split 507/508 showed comparable, if not better, performance when compared to split 565/566. The consistency in the best performing split sites across different CIDs effectively reduced the size of split site screening necessary for future designs with other inducible dimerization systems (Supplementary Fig. 9C, D). Through split site screening and rational optimization, we designed more than 4 split RfxCas13d ON-switch systems regulatable by GA, ABA, 4OHT, and Dano and a Dano-inhibited split Cas13d system with great inducibility (Fig. 3A).

We also aimed to design a Cas13 that can respond to endogenous signals, creating a dual-regulated Cas13 (small molecule and endogenous pathways). Such dual-regulated systems have proven to be invaluable for other genome engineering tools (e.g., recombinases) in animal model development[43,44]. As a proof of principle, we developed a WNT controllable system using the well-established SuperTOP Flash (STF) promoter to drive the expression of Cas13. This promoter responds to canonical WNT signaling involved in crucial cellular processes like fate determination, migration, and polarization[45] (Fig. 3B). First, we showed that we could achieve 72% knockdown in response to GSK3 inhibitor induction, an activator of the WNT pathway, when we used STF to control WTCas13d expression, suggesting a wide dynamic range in Cas13 expression level (Supplementary Fig. 10). Next, we replaced the WTCas13d with the GA-inducible split Cas13 gene cassettes and constructed an AND logic gate responsive to simultaneous exogenous and endogenous signaling. Results show that the AND logic circuit achieved efficient mCh knockdown only in the presence of both input signals. The basal mCh knockdown induced by the GSK3

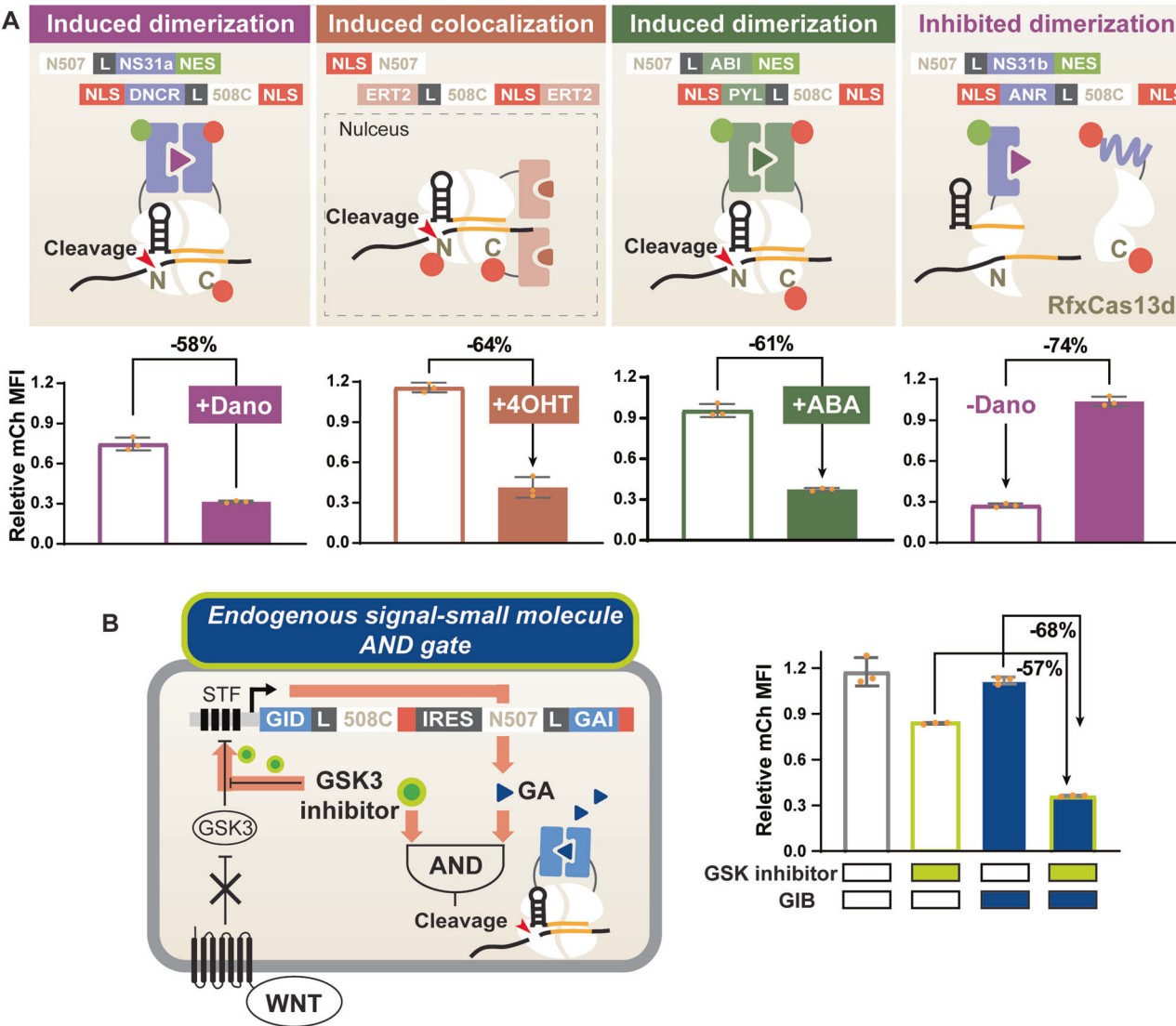

**Fig. 3 | Multiplex control of split Cas13d ribonucleases. A** The top panel illustrates 4 optimized split RfxCas13d designs responsive to Dano, 4-OHT, and ABA. The dNS3/DNCR domains dimerize upon Dano induction, resulting in ~60% mCh-specific knockdown. The ER[T2] domain localizes to the nucleus upon 4OHT induction, bringing the Cas13d C-terminal in close proximity with the N-terminal that is sequestered in the nucleus with an NLS, leading to ~60% induced mCh-specific knockdown with minimal leaky activity when uninduced. The ABI/PYL system dimerizes upon ABA induction, leading to 67% mCh-specific knockdown. While the former 3 systems are ON switches that become active upon induction, the dNS3/ANR system forms an OFF switch with split Cas13d. The dNS3/ANR system constitutively dimerizes until Dano competitively binds to dNS3 and displaces ANR, which inhibits Cas13d activity. Constitutive mCh-specific knockdown was over 70%, which is fully rescued with Dano induction. **B** Schematic of the small molecule-endogenous signal AND gate. A GSK inhibitor (green circles), mimicking the endogenous WNT signaling pathway, activates the WNT-responsive SuperTOP Flash (STF) promoter and activates the transcription of the GA-inducible split Cas13d. This Cascading transcriptional and post-translational control of split Cas13d achieved ~70% mCh-specific knockdown only in the presence of both the GSK inhibitor and GA (blue triangle). Data are presented as mean values +/− SEM. Error bars indicate the SEM for three biological replicates (*n* = 3). Source data are provided as a Source Data file.

inhibitor was not a result of leakiness from the inducible split Cas13d, as this mCh reduction also occurred when mCh targeting gRNA is absent (Supplementary Fig. 10), suggesting that the GSK3 inhibitor has a negative impact on the mCh expression independent of Cas13 activity.

## Multiplex inducible RNA regulation with orthogonal Cas13 effectors

The ability to independently control the expression of different genes (e.g., a genetic switchboard) is a highly desirable feature in an inducible gene regulatory platform as it enables the reprogramming of complex cellular functions and the recapitulation of natural cellular networks. As such, we developed inducible versions of different Cas13 orthologs. Among the discovered Cas13 orthologs, Cas13b derived from *Prevotella sp. P5-125* (PspCas13b) and Cas13a from *Leptotrichia wadei* (LwCas13a) enable specific and efficient RNA targeting activity in mammalian systems[29,31]. They have also been engineered for various applications such as RNA editing, imaging, splicing, and detection[29–31,33]. To first validate the orthogonality among Cas13a, b, and d orthologs, we transfected each Cas13 with its cognate gRNA and gRNAs of the other orthologs. Our results showed that mCh-specific knockdown only occurred when the effectors were transfected with their corresponding gRNAs (Supplementary Fig. 11).

To screen for active splits in PspCas13b and LwCas13a, we utilize the streamlined design process and insights gained from engineering split Cas13d. We predicted the PspCas13b and LwCas13a structures by aligning their sequences to their close family members whose structures have been resolved[46,47]. Then, we aligned their predicted

secondary structures with that of Cas13d[48]. Although Cas13b has a structure that is relatively different from the other members in the Cas13 family, we could still select sites that roughly align to active split sites in Cas13d for both PspCas13b and LwCas13a[46]. After screening with the GA-inducible dimerization domains, half (5 out of 10) of the selected sites for LwCas13a generated significantly induced knockdown (Supplementary Fig. 12A). Two sites in the HEPN1-I domain displayed induced knockdown levels similar to the wild-type, with the highest being a 50% decrease in mCh expression but only moderate inducibility (Supplementary Fig. 12B).

In comparison, the screening of Cas13b splits generated multiple systems with larger than 0.4 inducibility (Supplementary Fig. 13B). Interestingly, 3 of the 12 screened split sites in PspCas13b aligned with RfxCas13d split sites that generated robust induced knockdown consistently with different CID systems, and so did the active split sites in LwCas13a (Supplementary Table 3). These synchronized hits are localized within the HEPN1 domains[40,46,47]. Split 49/50 in PspCas13b and split 416/417 and 421/422 in LwCas13a were selected due to their alignment to the RfxCas13d split site 177/178 and 180/181 (Supplementary Table 3). Split 250/251 in PspCas13b and split 786/787 in LwCas13a were selected due to their alignment to the RfxCas13 split site 507/508 (Supplementary Table 3). The recurring active splits in the HEPN1 domain implied that, potentially, this is a functionally or structurally conserved region in the Cas13 family. Moreover, moving from designing RfxCas13d to PspCas13b, the number of screened split sites has been reduced by more than half, which greatly reduced the effort in this design step and suggests the transferability of design insights across Cas13 family members (Fig. 1D, Supplementary Figs. 12B, 13B)

Besides split site activity, the CID orientation preference is also transferred to split PspCas13b (Supplementary Fig. 13C). We found that the orientation N-GAI/GID-C again generated performance closer to the ideal than the original orientation with 4 out of 5 splits (Supplementary Fig. 13C), suggesting the extensibility of the design rule we gained from inducible split Cas13d. To further increase the inducibility of this system, we redesigned the system with different numbers and locations of the NLSs and NESs (Supplementary Fig. 13D). We confirmed that targeting split moieties to different compartments led to higher inducibility (Supplementary Fig. 13D). Through the same workflow, we observed consistency in active Cas13b split sites across all CIDs and preferred CID orientation from split Cas13d to b, and similar effectiveness of NES/NLS tagging for rescuing leakiness. With these design rules, we were able to design ABA-inducible, Dano-inducible, and Dano-inhibited split Cas13b systems with inducibility around 0.7, using 45% fewer screening constructs (Fig. 4A, Supplementary Fig. 14).

Finally, to demonstrate the complete orthogonality of the inducible Cas13d and Cas13b, we transfected cells with both the GA-inducible Cas13d and Dano-inducible Cas13b systems targeting B2M and mCh and added either no drug, each drug individually or both (Fig. 4B). Each inducible system successfully knocked down its specific target in the presence of its corresponding inducer with no crosstalk, as shown by the intact expression level of the other target (Fig. 4B). Flow cytometry data also showed that in the presence of both GA and Dano, the same population of cells had reduced mCh and HLA expression simultaneously (Supplementary Fig. 15A). Besides the GA and Dano-inducible system, our library offered at least 2 other combinations of orthogonal control, GA with ABA, and ABA with Dano (Supplementary Fig. 15C, D).

While we can achieve multiplex regulation, we can also demultiplex small molecule control, which divergently regulates more than one output with a single input (Fig. 4C). We utilized the Dano-inducible and Dano-inhibited systems, where they respond to Dano simultaneously, and regulate 2 genes in the opposite mode: in the absence of Dano, Dano-inhibited Cas13d knocks down B2M expression, while Dano-inducible Cas13b is inactive and mCh expression is intact; in the presence of Dano, the activity of Cas13b and d is reversed, and B2M is free from Cas13d mediated downregulation while mCh is targeted for knockdown by Cas13b (Fig. 4C, Supplementary Fig. 15B).

## RNA-level incoherent feed-forward loop (IFFL) to achieve adaptation response

Synthetic feedback circuits are critical for engineering advanced functions in living cells. In particular, the incoherent feed-forward loop (IFFL) represents an important class of network motif that is found in diverse cellular systems from all kingdoms of life, including mammalian cells[49]. Certain types of IFFLs can lead to a pulse generator−transient output signals in response to a sustained input signal, and enable adaptation and homeostasis, which is critical in many signal transduction or sensory networks[50,51]. To achieve a synthetic adaptation response that operates at the post-transcription (RNA) level, we constructed an IFFL utilizing the Dano-inhibited Cas13b OFF switch (Fig. 5A). Our circuit comprises an activation arm and a delayed repression arm (Fig. 5A). The Dano-inhibited Cas13b OFF switch serves as the activation arm that represses, in the absence of Dano, the output mCh and the repression arm effector, WT Cas13d (Fig. 5A). Upon Dano induction, the repression is lifted to express mCh and WTCas13d. WT Cas13d serves as the delayed repression arm that represses output expression after its expression accumulates (Fig. 5A). We engineered the WT Cas13 to be expressed along with a GFP to monitor the expression of this intermediate step (Fig. 5A). In addition, we used mCh fused with the destabilization domain of FKBP12 (DD-FKBP12) as the output to track the expression dynamics.

We transfected HEK293FT cells with either the complete IFFL circuit or the same circuit missing the repression arm. We tracked the dynamics of the circuit output after the induction of the same population of cells using live cell fluorescence imaging. Results show that only the mCh output generated a pulse pattern in response to Dano, while the GFP repression arm expression continues to increase over time (Fig. 5B). Furthermore, the pulse characteristics can be regulated by tuning Dano concentration. Higher Dano concentration leads to a pulse peak with a short induction time and faster adaptation (Fig. 5C, Supplementary Fig. 16C). To independently confirm the dynamics of our circuit, we also used flow cytometry to measure the output expression in response to different duration of induction (Supplementary Fig. 16). Similar to live cell fluorescence imaging, we show that only the complete IFFL circuit responds to Dano induction with an adaptation response longer than 12 h (Supplementary Fig. 16B). By 48 h after induction, output expression is comparable to the uninduced condition (Supplementary Fig. 16B). Without the repression arm, output expression increases and plateaus after 24 h of induction with no adaptation (Supplementary Fig. 16B).

To decipher the contribution of the repression arm in circuit performance, we replaced the WTCas13d with a GA-inducible Cas13d whose activity is tunable by varying GA concentration (Supplementary Fig. 16D). Results show observed pulse pattern is regulated by both Dano and GA, in peak amplitude, peak timing, and adaptation efficiency (Fig. 5D−F Supplementary Fig. 16E). Our circuits will ultimately allow a "plug-and-play" tunable adaptation of any endogenous gene as an output in response to the input signal.

## Chemical-inducible Cas13-mediated gene expression knockdown in mice

To demonstrate the in vivo multiplex performance of our inducible Cas13 platform, we applied the GA-inducible Cas13d and ABA-inducible Cas13b simultaneously in mice (Fig. 6A). We deployed a dual-bioluminescence reporter system with Antares and firefly luciferase (Fluc) as the targets (Fig. 6A). Both luciferases allow non-invasive signal detection in vivo, and their orthogonal substrates enable separated signal detection within the same animal[52]. We introduced Balb/c mice with plasmids carrying the inducible split systems, luciferase-targeting

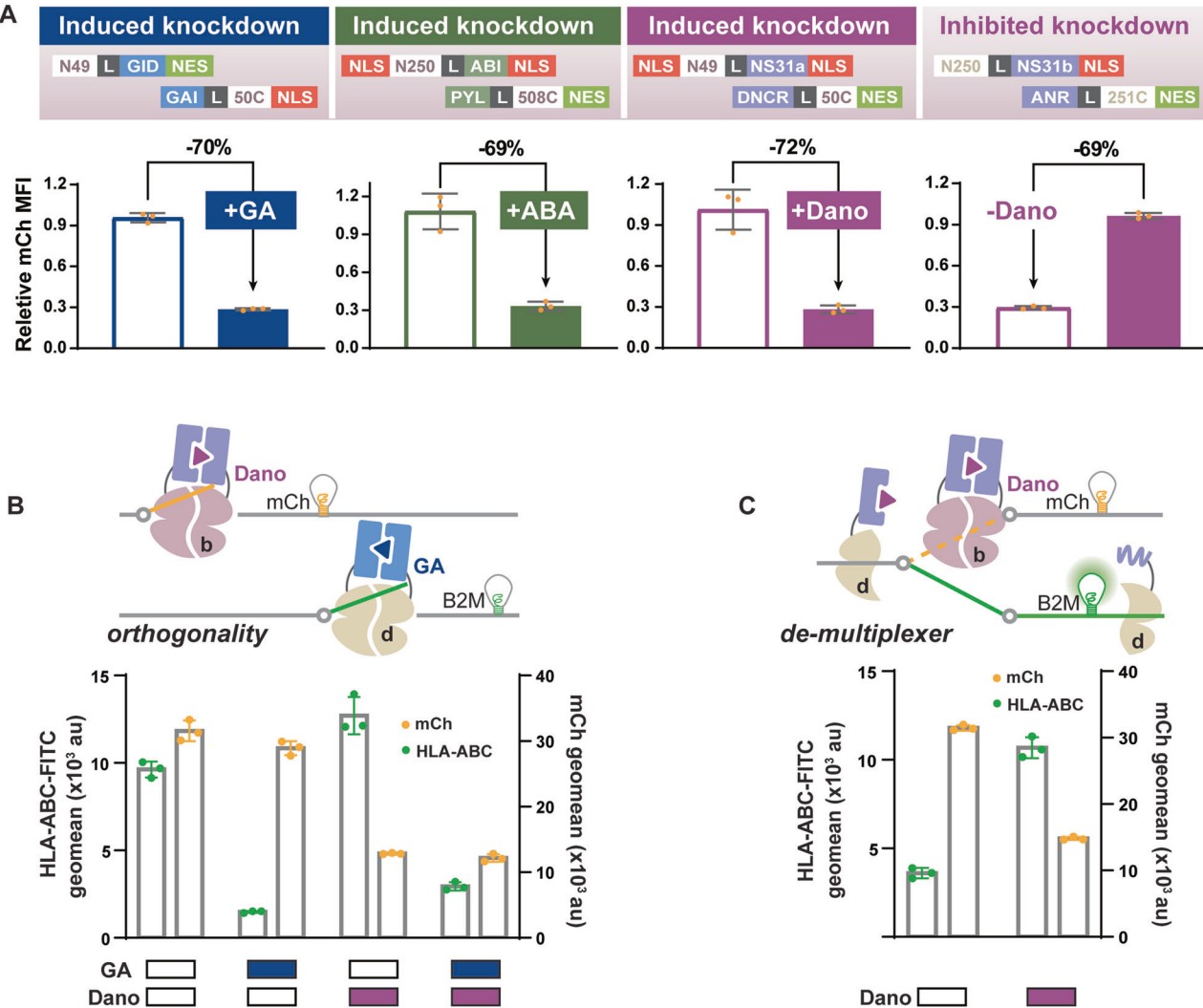

Fig. 4 | Orthogonal multiplex inducible RNA regulation. A Collection of optimized small molecule-inducible split Cas13b systems responsive to GA, ABA, and Dano. All ON and OFF switches with Cas13b generate mCh-specific knockdown with a dynamic range of around 70% between induced and uninduced conditions and no leaky activity in the OFF state. B Dano-inducible Cas13b regulates mCh expression, while GA-inducible Cas13d orthogonally regulates HLA surface expression through targeted knockdown of B2M. Results show ~60% reduced mCh expression only upon Dano induction, while HLA expression is downregulated by >80% with GA induction. No cross-talk between the GA-inducible Cas13d and Dano-inducible

Cas13b was observed. Dual knockdown of mCh and HLA upon simultaneous stimulation of GA and Dano is as efficient as the knockdown when induced separately. C Dano-inducible Cas13b regulates mCh expression while Dano-inhibited Cas13d regulates HLA surface knockdown. Upon Dano induction, Cas13b is actively knocking down mCh, while Cas13d is inactive. In the absence of Dano, Cas13d inhibits HLA expression while Cas13b is inactive, and mCh is highly expressed. Data are presented as mean values +/− SEM. Error bars indicate the SEM for three biological replicates (n = 3). Source data are provided as a Source Data file.

gRNAs, and luciferases via hydrodynamics transfection. IVIS imaging after either 2 doses of inducer or vehicle revealed that luminescence was only reduced in mice with both target gRNAs and the corresponding small molecules (Fig. 6A–C). The GA-inducible Cas13d system with the Fluc targeting gRNA achieved 98% knockdown of Fluc with GA induction, while expression and dimerization of split Cas13 effectors alone did not change reporter expression without the targeting gRNA (Fig. 6B). The ABA-inducible Cas13, in contrast, knocked down Antares expression by 65% with ABA induction (Fig. 6C). The reduced efficiency compared to in vitro results is likely due to the less concentrated ABA dosage we used in vivo. Additionally, neither GA nor ABA-inducible Cas13 in the uninduced conditions generated target knockdown with gRNAs, suggesting the in vitro optimizations are transferable in vivo.

After validating the induced knockdown by each inducible system individually, we transfected mice with both systems to achieve *simultaneous* regulation of the 2 targets (Fig. 6D). IVIS imaging after dosing

of vehicle control, single small molecule, or both molecules shows that target-specific knockdown was orthogonally induced by the corresponding small molecule, suggesting the complete orthogonality between both inducers and Cas13 effectors for the GA and ABA-inducible systems in vivo (Fig. 6D). Meanwhile, knockdown by each of the inducible Cas13 effectors is as efficient as when they were applied individually: GA-inducible Cas13d achieved 86% and 88% induced knockdown in GA and dual induction, respectively, and ABA-inducible Cas13b mediated knockdown was 63% and 71% with ABA and dual induction (Fig. 6D). In these experiments, we transfected system components in individual plasmids to adjust component ratios for optimal performances. However, compact plasmid design can be deployed for more efficient delivery of more complex circuits.

## Discussion
We developed our CRISTAL platform driven by a powerful and versatile collection of 10 inducible split Cas13 ribonucleases and

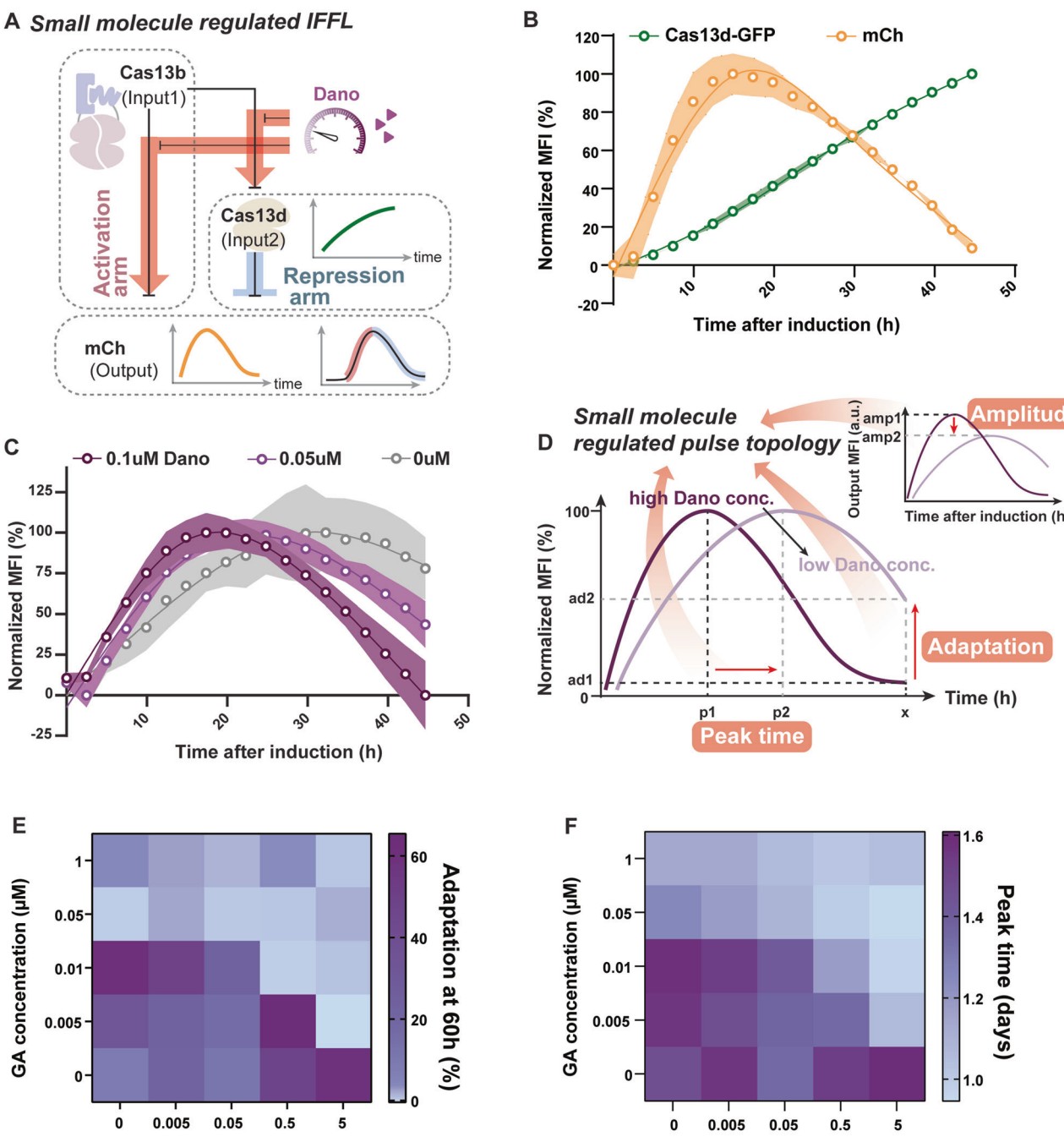

**Fig. 5 | Cas13-based RNA-level IFFL pulse generator. A** The incoherent feed-forward loop, IFFL, is composed of an activation and a repression arm, where the activation arm, Dano-inhibited Cas13b, regulates both the output mCh and the repression arm Cas13d which also regulates output expression. In response to a sustained Dano-induction, output expression increases transiently and then undergoes adaptation, while the repression arm expression, GFP, consistently increases. **B** HEK293FT cells were transfected with the entire circuit, induced with Dano, and imaged for mCh and GFP mean fluorescent intensity (MFI) over time. As Cas13d-GFP expression consistently increased after Dano-induction, mCh adapted back to the basal expression level in response to the sustained Dano induction. Data are presented as mean values +/− SEM. Shaded area indicates the SEM for three biological replicates (n = 3). **C** Reducing Dano concentrations resulted in reduced adaptation responses of the circuit. Output expression normalized to mode is shown here, raw output expression data is shown in Supplementary Fig. 16C. Data are presented as mean values +/− SEM. Shaded area indicates the SEM for three biological replicates (n = 3). **D** A schematic showing different pulse topologies generated with varied small molecule concentrations regulating the activation arm.

Parameters, such as the peak time, adaptation, and amplitude, can be used to quantitatively describe the pulse topology: higher concentrations of Dano, a stronger activation arm, may lead to faster peak time, increased peak amplitude, and decreased adaptation efficiency. **E** When we replace the repression arm of the IFFL with the GA-inducible Cas13d, the output pulse topology becomes susceptible to changes in both the activation regulator, Dano, and the repression arm regulator, GA. Heatmap shows the adaptation of output mCh at 60 h for different Dano and GA concentrations. With 5 uM Dano induction, output achieves perfect adaptation (~0%) at 60 h with minimal GA induction, as the repression arm is also expressed strongly along with the output. With a decreasing Dano concentration, a higher GA concentration is required for perfect adaptation. Experiment was repeated 3 times, and a representative experiment is displayed. **F** Peak time is affected by both Dano and GA in the dual-regulated IFFL. Output expression peaks sooner and starts to decrease in a short time with simultaneous strong induction of Dano and GA. Experiment was repeated 3 times, and a representative experiment is displayed. Source data are provided as a Source Data file.

**A  IN VIVO EXPERIMENTAL DESIGN**

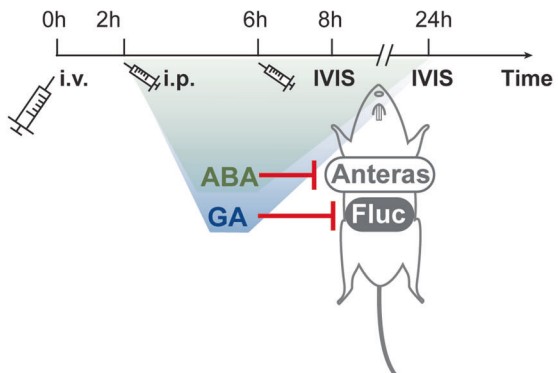

**B  GA INDUCED FLUC KNOCKDOWN**

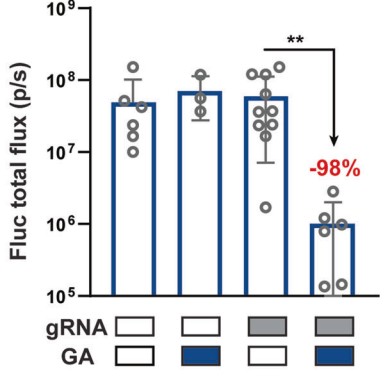

**C  ABA INDUCED ANTERAS KNOCKDOWN**

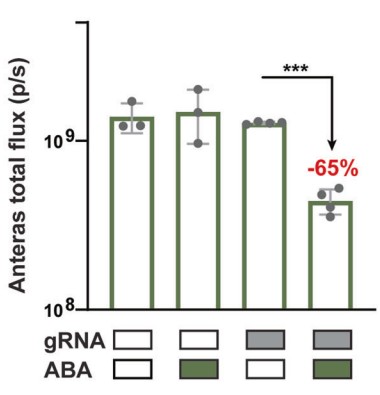

**D  ORTHOGONAL AND MULTIPLEXED KNOCKDOWN**

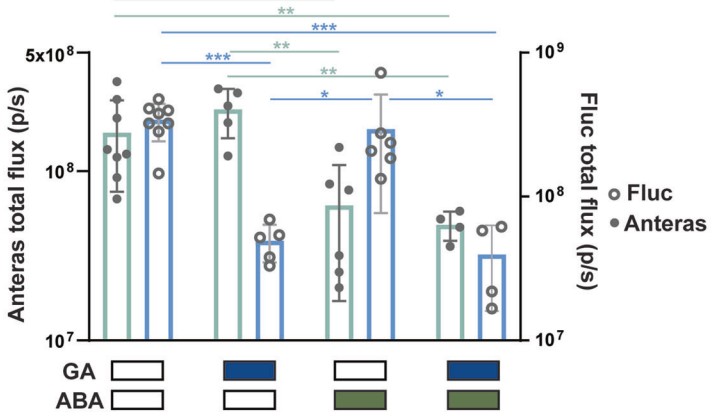

**Fig. 6 | Simultaneous and orthogonal regulated gene knockdown in mice.**
**A** Experimental timeline of in vivo demonstration of GA and ABA-inducible knockdown of Fluc and Antares, respectively by Cas13d and Cas13b. Plasmids carrying the inducible split Cas13 effectors, gRNAs, and the target luciferase are delivered in vivo using hydrodynamic tailed vein transfection at 0 h. Inducers were injected intraperitoneally at 2 h and 6 h after transfection. In vivo luminescence imaging was performed 8 h after transfection for Fluc expression and 24 h after transfection for Antares expression. i.v., intravenous; i.p., intraperitoneal. **B** The GA-inducible Cas13d generated 98% reduction in Fluc luminescence in the GA-inducible targeted group compared to uninduced group in vivo. Sample replicates left to right: $n = 6$, $n = 3$, $n = 10$, $n = 6$; $P = 0.00641$. **C** The ABA-inducible Cas13b generated 65% reduction in Antares luminescence in the ABA-inducible targeted

group compared to uninduced group in vivo. Sample replicates left to right: $n = 3$, $n = 3$, $n = 4$, $n = 4$; $P = 8.56E-5$. **D** Mice were transfected with both Fluc targeting GA-inducible Cas13d system and Antares targeting ABA-inducible Cas13b system, randomly injected with either vehicle control, GA, ABA, or GA and ABA. Luminescence imaging shows that Fluc and Antares were orthogonally regulated by GA and ABA in vivo. Sample replicates left to right: $n = 8$, $n = 5$, $n = 6$, $n = 4$; $P$-values from top to bottom: $p = 0.0177$, $p = 0.00788$, $p = 2.93E-05$, $p = 0.00397$, $p = 5.64E-05$, $p = 0.00504$, p(left) = 0.0397, p(right) = 0.0346. $P$-values were calculated as a 2-tailed $t$-test. Data are presented as mean values +/− SEM. Error bars indicate the SEM for at least three biological replicates ($n \geq 3$). Source data are provided as a Source Data file.

demonstrated their compatibility and orthogonality. Our CRISTAL platform enables us to execute multiplex inducible post-transcriptional regulation in mammalian cell culture and mice. We also show that our system is surprisingly robust for engineering complex gene circuits, such as an AND logic gate, parallel multiplex and demultiplex circuits, and an IFFL.

A key component of our design is the RNA targeting effector module. We took advantage of the recently discovered CRISPR/Cas13 system as an RNA regulatory module[28,29,31]. Although other CRISPR systems, like CRISPR/Cas9, can regulate RNA transcription[12] and redirect to target RNA[53], their targeting abilities are still restricted due to closed chromatin structure, specificity against gene isoforms, and a PAM sequence requirement. In contrast, Cas13 inherently has higher targeting specificity and wider targeting range, especially against unconventional RNA types such as circular RNAs and lncRNAs[23,28].

We also found that many functional split sites in Cas13a, b, and d are structurally aligned. These alignments reduced the number of split

sites screened by 2-fold from the first attempt of splitting RfxCas13d to the second in PspCas13b and LwCas13a. This suggests hot spots of inducible split sites among Cas13 family members[54–56]. As the number of new members within the CRISPR/Cas13 family increases, the design pipeline we developed here will aid the engineering of these new effectors. However, as we are disturbing the Cas13 structure, further characterization of gRNA array processing ability, gRNA mismatch tolerance, off-target effect, and collateral cleavage activity of our inducible systems is needed.

In light of these considerations, we conducted studies addressing the issue of collateral RNA cleavage. While initial assessments suggested limited collateral activity in eukaryotic cells[28,29,31], concerns regarding Cas13-linked collateral activity have become more prominent[54,57–59]. We systematically investigated collateral activity and focused on the comparison between inducible split systems and their WT counterparts (Supplementary Fig. 17). Notably, split Cas13 systems display comparable or heightened specificity (reduced collateral activity) across cell types, including U87 cells where the WT Cas13

effectors demonstrate extensive collateral activity[57,60] (Supplementary Fig. 17). Additionally, we establish that PspCas13b induces less collateral activity compared to RfxCas13d in HEK, U87, and Neuro2A cells (Supplementary Fig. 17A–C).

Apart from transgene reporter-based experiments, we also monitored exogenous and endogenous gene expression using split Cas13 systems for diverse targets (Figs. 2C, 4C, D, and Supplementary Fig. 5D–F). Our findings align with recurring patterns observed in previous collateral activity studies: collateral activity intensifies when Cas13 targets highly expressed transcripts (Supplementary Fig. 18C, D)[58,59] and collateral cleavage of exogenous genes is more profound[58–60]. Conversely, discrepancies in the WT Cas13-associated collateral activity across various models and conditions stem from factors like cell types, target transcripts, Cas13 orthologs, and gRNA choices[54,57–60]. This inconsistency, however, suggests opportunities to mitigate collateral activity and its associated risks through careful design strategies for gRNA, the selection of suitable orthologs, and targeting relatively low-expression level genes[61]. More importantly, inducible split PspCas13b consistently shows minimal collateral activity, although inducible split Cas13d CRISTAL systems proposed in our manuscript show reduced but still significant collateral activity compared to WT systems (Supplementary Fig. 17A), the tunable activity of the CRISTAL platform enhances our ability to effectively manage collateral activity: collateral activity decreases with on-target activity (Supplementary Fig. 18).

In addition to its RNA cleavage activity, Cas13 effectors possess RNA binding abilities that have been harnessed to facilitate a range of functions such as programmable RNA tracking, modification, editing, and the study of RNA-protein interactions in mammalian cells[31,33,34,62]. Recent advancements have demonstrated the possibility of conditional RNA modification using an ABA-inducible split Cas13b system[63]. This system allows for the site-specific and time-dependent introduction of m6A writing on RNAs using a fused methyltransferase-like 3 enzyme[63]. The inducible dimerization of the catalytically dead split Cas13b also allows for the controlled release of Cas13 to prevent unwanted perturbation of the targeted transcripts[63]. The CRISTAL platform offers various inducible split Cas13 systems that not only silence RNA with great efficacy, but also have the potential to be coupled with corresponding functional domains to achieve conditional, reversible, and multiplex RNA manipulations.

We anticipate that using CID systems as the exogenous control module in our design will facilitate clinical translation. For example, CID systems are employed in Chimeric Antigen Receptor (CAR) T cells as safety control switches[64]. We also did not observe variation in the activity of the 2 modules across contexts (in different cell types or in vivo), suggesting robust translation among different models. The only reduced activity was observed in primary human T cells, which is probably due to inefficient gene delivery. The availability of orthogonal CIDs and Cas13 effectors allows us to independently regulate multiple genes simultaneously in the same cells. Furthermore, additional CID systems have emerged through genome mining and de novo protein design[37–39]. These distinct CIDs, along with our design rules and well-cauterized split sites, will facilitate further expansion of our CRISTAL platform. Our collection of split Cas13 is also compatible with light-inducible domains to grant spatiotemporal control, and protein-regulated dimerization systems to rewire gene expression to Cas13 activity[65–67]. The ability to dynamically regulate multiple transgenes and/or endogenous genes independently in the same cell enabled by our CRISTAL platform is very powerful and will help reveal the temporal order of dynamic changes in various important and complex cellular processes, such as cell differentiation and cancer metastasis.

An important benefit of designing post-translational regulation of Cas13 activity is that it can be coupled with other modes of regulation, such as transcriptional control. For instance, Cas13 fragments can be combinatorially regulated by multiple promoters to achieve cell state-specific Cas13 expression and activity. We have demonstrated the feasibility of using the WNT pathway responsive promoter for the controlled expression of WT Cas13 and inducible Cas13 systems. Other than transcriptional control, our system can also be layered with translational control, such as with the recently developed ADAR-based RNA sensing system, and create a sense-and-action device responsive to RNA input and output also transcriptome modulations[15,68].

A primary goal in synthetic biology is to predictably engineer cellular networks, with a demonstrated impact in understanding fundamental biology and medicine[69]. Due to the orthogonality of Cas13, and the low leakiness and high dynamic range for some of our inducible Cas13 effectors, we were able to engineer a robust and tunable IFFL, a foundational network motif with complex dynamics widely found in various cellular functions. The perfect adaptation offered by our IFFL resembles the homeostasis response, which is particularly important in sensory neuron activation and cellular mitotic decision-making[50,51,70]. Moreover, our CRISTAL platform can seamlessly regulate endogenous transcripts simply by changing the gRNA and allow us to readily generate complex endogenous gene expression profiles.

The robust and specific activity of the inducible Cas13 splits against highly expressed transgene targets in mice also suggests sufficient cleavage activity toward endogenous RNA targets. Furthermore, multiplexed regulation unleashes the potential for constructing complex circuits in vivo. Although our experiments involve only transient expression of the inducible systems, stable integration methods such as the PiggyBac transposon system may allow us to generate animal models with sustained system expression and activity[71]. Together, our CRISTAL platform has the potential to facilitate complex circuit engineering in vivo with custom-designed transcriptome profiles for basic studies and therapeutic applications.

## Methods
### Ethical statement
Animal studies were conducted at the Boston University Medical School Animal Science Center under a protocol (PROTO201800600) approved by the Boston University Institutional Animal Care and Use Committee. All animal experiments were performed following the relevant institutional and national guidelines and regulations. Human blood collection was performed under a protocol (#18-1537) approved by the Institutional Review Board (IRB) at Boston University.

### Human Embryonic Kidney (HEK) cells
HEK293FT cells were cultured at 37 °C and 5% $CO_2$ in Dulbecco's modified Eagle's medium (DMEM) containing 5% fetal bovine serum (FBS; Thermo Fisher 10437028), 50 UI/ml penicillin/streptomycin (Corning 30001CI), L-glutamine (Corning 25005CI) and 1 mM sodium pyruvate (Lonza 13115E).

### Human T lymphocyte (Jurkat)
Jurkat clone E6-1 (ATCC TIB-152) was maintained in RPMI 1640 (Lonza 12-702Q) with L-glutamine supplemented with 5% FBS, 100U Penicillin/Streptomycin and an additional 2 mM L-glutamine at 37 °C and 5% $CO_2$. Cells were passaged 1:4 every two days with a complete media change.

### Mouse neuroblast (Neuro-2a)
Neuro-2a (ATCC CCL-131) cells were maintained in Eagle's Minimum Essential Medium (EMEM Corning 10-009-CV) supplemented with 10% FBS and 100U Penicillin/Streptomycin at 37 °C and 5% $CO_2$. The cells were passaged 1:3 every 3 days reaching 70–80% confluency.

### U87MG (U87)
U87MG (ATCC HTB-14) cells were maintained in Eagle's Minimum Essential Medium (EMEM Corning 10-009-CV) supplemented with 10% FBS and 100U Penicillin/Streptomycin at 37 °C and 5% $CO_2$. The cells were passaged 1:5 every three days reaching 70–80% confluency.

## Primary human T cell isolation and culture

Blood samples were obtained from the Blood Donor Center at Boston Children's Hospital (Boston, MA). Primary peripheral blood mononuclear cells (PBMCs) were isolated using Lymphoprep density medium using the manufacturer's protocol. Isolated PBMCs were either maintained fresh or stored as cell stocks in liquid nitrogen until needed. PBMCs were maintained in X-Vivo 15 media (Lonza 04-418Q) supplemented with 5% human AB serum (Valley Biomedical HP1022), 10mM N-acetyl L-Cysteine (Sigma A9165), 55 µM 2-Mercaptoethanol (Thermo Fisher 21985023), and 50–100 U/ml IL-2 (NCIBRB Preclinical Repository), with a concentration between 500,000 cells/ml to 1,000,000 cells/ml.

## Animals

Female Balb/c mice, 5–6 weeks old, were purchased from Jackson Laboratories (JAX 00651) and used for in vivo induced knockdown experiments. For all experiments, mice were randomly assigned to experimental groups.

## Protein structural analysis and identification

HHPred[72] was utilized to predict secondary structures for RfxCas13d, LwCas13a, and PspCas13b using crystal structure data for EsCas13d (PDB:6E9E), LbaCas13a (PDB:5XWP), and PbuCas13b (PDB:6DTD). Secondary structure alignments for EsCas13d, LbaCas13a, and PbuCas13b were adapted using the I-TASSER server for protein structure and function prediction[48].

## Inducible split Cas13 DNA assembly

Split Cas13 fragments were cloned into a series of sub-cloning vectors containing CID domains assembled previously for split recombinase systems. Cas13 N-terminal fragments were first PCR amplified out of a full Cas13 sequence template, pXR001: EF1a-CasRx-2A-EGFP (Plasmid #109049), pC0046-EF1a-PspCas13b-NES-HIV (Plasmid #103862), and pC014 - LwCas13a-msfGFP (Plasmid #91902). Each contained a 5′ sequence to code for a MluI restriction site, a Kozak consensus sequence, a 3′ sequence to code for a BspEI restriction site, a glycine-serine-rich linker L1 (SGGSGSGSSGGSG), and a KpnI restriction site. Through Gibson reactions, the purified PCR product from a verified band size (Epoch Life Science) was inserted into a gel-purified N-terminal sub-cloning vector digested with MluI and KpnI restriction enzymes (New England Biolabs). Cas13 C-terminal fragments were also PCR amplified out of the full Cas13 sequence templates mentioned above to contain a 5′ sequence to code for the KpnI restriction site a start codon, a 3′ sequence to encode a stop codon, and an EcoRI restriction site. Through Gibson reactions, the purified PCR product from a verified band size (Epoch Life Science) was inserted into the gel-purified C-terminal sub-cloning vector digested with EcoRI and KpnI restriction enzymes (New England Biolabs). Reaction products were transformed into chemically competent Top10 *Escherichia coli* cells and selected on LB-agar plates with carbenicillin at 37 °C static conditions. Colonies were picked the following day, inoculated into carbenicillin-containing LB medium, and grown in a 37 °C shaking incubator (Infors) for 16–20 h. Plasmid DNA was extracted using a mini plasmid preparation kit (Epoch Life Science). Analytical digests with MluI, KpnI, and EcoRI were then performed and run on gel electrophoresis to assay whether a correct product was made. Plasmid clones with correct gel bands were sent for sequencing (Quintara Biosciences).

## Small-molecule preparation

1000x stocks of ABA (100 mM, Gold Biotechnology A-050-500) and GA (10 mM, Toronto Research Chemicals G377500) were prepared by dissolving in 100% ethanol and stored at −20 °C. 10000x stocks of Dano (10 mM, MedChem Express HW-10238) and CHIR 90021 (GSK3 inhibitor) (10 mM, Tocris Bio-Techne 4423) were prepared by

dissolving in dimethyl sulfoxide (DMSO) and stored at −20 °C. 50000x stocks of 4OHT (50 mM HelloBio HB2508) were prepared by dissolving in DMSO and stored at −20 °C. For in vitro induction, the stock solutions were diluted to 5X with culture media and then added to the cell cultures such that the final induction concentration is 1X. For in vivo administration, GA was dissolved to 2 mg/ml in 2.5% DMSO/PBS and ABA was dissolved to 20 mg/ml in 2.5% DMSO/corn oil (MedChemExpress HY-Y1888. Mice received 100ul inducer for each induction.

## Human Embryonic Kidney (HEK) cells transfections

Polyethylenimine (PEI) (Polysciences 23966-2) was used to transfect HEK293FT cells. PEI was dissolved in PBS to a concentration of 0.323 g/L and pH was adjusted to 7.5. HEK293FT cells were plated at 200,000 cells/ml in tissue culture-treated plates 1 day before transfection such that they were 60%-80% confluent on the day of transfection. For 96 well plates, each well was plated with 20,000 cells on day 0 and transfected with 100 ng plasmid DNA (0.8ul PEI solution and 0.15 M NaCl solution up to 10 ul total) on day 1. HEK293FT cells were induced with either vehicle control or 10 uM GA, 1uM Dano, 100 uM ABA, and 1 uM 4OHT., Transfection mixtures were prepared as master mixes for replicates and incubated for 15–20 min at room temperature before being added to cell culture.

## Human T lymphocyte (Jurkat) transfections

For Jurkat transient transfection, 20 million Jurkat cells at 0.6–0.8 million/ml density were mixed with 5 ug of a transfection marker (iRFP), 5 ug of a target gene (mCh), 10ug of each split Cas13d and 10 ug of gRNA plasmid (total 40 ug DNA per transfection) in 300 ul of a transfection medium (Jurkat complete medium without antibiotics) and incubated for 15 min. Cell/DNA mixture was then transferred to a 4-mm gap width electroporation cuvette (Fisher Scientific FB104 or Lee Plastic Company 4040-04) and electroporated using a BTX electroporator (Harvard Apparatus BTX Electro Square Porator ECM 830). Electroporation conditions were square pulse, 300 V, 10 ms pulse length, and single pulse. Shocked cells were incubated for 10 min and transferred to a 6-well with 10 ml of the transfection medium for 2 h of recovery incubation. For induction experiments, transfected cells were plated in 96-well plates with >3 replicates and induced with either vehicle control or 5 uM GA, 100 nM Dano, 50 uM ABA, and 500 nM 4OHT.

## Mouse neuroblast (Neuro-2a) transfections

For Neuro-2a transient transfection, 50,000 Neuro2a cells in 250ul complete medium were plated per well in 48-well plates 24 h before transfection. After a media change on the day of transfection, Neuro-2a cells (60-80% of confluency) were transfected with 31.25 ng of a transfection marker (iRFP), 31.25 ng of a target gene (mCh), 62.5 ng of each split Cas13d and 62.5 ng of gRNA plasmid (total 250 ng DNA per well) using Lipofectamine3000 (Thermo Scientific L3000008) according to the manufacturer's protocol (DNA: Liposome = 2:3 w/v). Inducers (5 uM GA, 50 nM Dano, and 50 uM ABA) or vehicle control were added 2 h after transfection.

## U87MG (U87) transfections

For U87 transient transfection, 20,000 U87 cells in 100 ul complete medium were plated per well in 96-well plates 24 h before transfection. After changing complete medium to Opti-MEM reduced serum medium (Fisher Scientific 31-985-062) on the day of transfection, U87 cells (60–80% of confluency) were transfected with 10 ng of a transfection marker (iRFP), 10 ng of the collateral activity reporter gene (GFP), 20, 2, 0.2 or 0 ng of the target gene (mCh), 20 ng of each split Cas13d and 20 ng of gRNA plasmid (total 100 ng DNA per well) using Lipofectamine3000 according to the manufacturer's protocol (DNA: Liposome = 2:3 w/v). Inducers or vehicle control were added 6 h after transfection with a medium change.

## Endogenous gene expression analysis

Cells harvested 48 h after induction for RNA extractions. Cell pellets collected from 48-well plates were lysed and total RNAs were extracted using the Direct-zolTM RNA Miniprep kit (Zymo Research R2053) according to the manufacturer's protocol. 500 ng–1 ug of total RNA was reverse-transcribed in a 20 ul reaction using the qScript cDNA SuperMix (Quantabio 95048-100) according to the manufacturer's protocol (5 min at 25 °C, 30 min at 42 °C, and 5 min at 85 °C). The resulting cDNA was diluted 1 to 5 and 10–20 ng of cDNA, assuming full conversion, was amplified in a 5 ul qPCR reaction using x2 LightCycler 480 SYBR Green 1 Master mix (Roche Diagnostics 04707516001) according to the manufacturer's protocol (15 s at 95 °C and 30 s at 60 °C, 40 cycles). RTqPCR reactions were run in a 384-well format with a LightCycler 480 Instrument II (Roche). RTqPCR primers (IDT) for SYBR Green are listed in Supplementary Table 5. GAPDH was used as the reference gene and relative gene expression levels were analyzed by the ddCt method.

## Stable cell line generation and reversibility experiment

Two expression cassettes each carrying a CAG promoter driving the expression of NCas13d-GAI-T2A-BFP and GID-Cas13dC-T2A-iRFP were packed into a single PiggyBac-transposon vector plasmid (System Biosciences). HEK293FT cells were transfected with the vector and transposon plasmid (System Biosciences). Transfected cells were incubated for 72 h before selection with media containing 2 ug/ml Puromycin (Invivogen ant-pr-1). Cells were selected for at least 3 passages before fluorescence activated cell sorting (FACS). Single clones that highly expressed GA-inducible split Cas13d (iRFP and BFP) were sorted, expanded, and transduced with concentrated lentivirus carrying either a non-target gRNA or a B2M targeting gRNA expression cassette and a GFP expression cassette for verifying successful integration.

On day 0 of the reversibility experiment, a stable cell line was plated in 96-well plates, and triplicates were either uninduced (group d) or induced with GA (group a) (Supplementary Table 6). Group b and c were induced on days 1 and 2, respectively. On day 3, cells were sampled for surface HLA staining and flow cytometry (at least 10,000 cells per sample). Data collected on day 3 is HLA and fluorescent protein expression with 0 (d), 1 (c), 2 (b), and 3 (a) days of GA induction. Groups b, c, and d were passaged with GA supplement, while group a was passaged with normal media. On days 4 and 5, GA was washed out of groups b and c, respectively, to start recovery from GA induction. On day 6, cells were sampled for surface HLA staining and flow cytometry (at least 10,000 cells per sample). At this time point, groups a, b, and c recovered, respectively, 3, 2, and 1 days from 3 days of GA induction. Group d had been induced for 3 days. Cells were passaged with no GA media after sampling for flow cytometry. On day 9, cells were sampled for surface HLA staining and flow cytometry (at least 10,000 cells per sample). Groups a, b, c, and d recovered 6, 5, 4, and 3 days respectively after 3 days of induction. After sampling, all groups were passaged. Group a received the second round of GA induction, and groups b, c, and d were passaged with no GA media. On day 10, group b received the second round of GA induction. On day 11, all samples were collected, and flow cytometry was run with groups a and b receiving 2 and 1 day of second GA induction. (Supplementary Table 6)

## Fluorescence Activated Cell Sorting (FACS)

FACS was performed for HEK293FT cell lines stably integrated with GA-inducible split Cas13d cassettes on an SH800 Cell Sorter (Sony Corporation). Cell samples were suspended in 2 ml 1X Phosphate Buffered Saline containing 1% Fetal Bovine Serum and passed through a 0.45 um filter to break clumps. Live cells were gated by the forward scatter (FSC) and the side scatter (SSC). Fluorescence data was collected for BFP (excitation laser: 405 nm, emission: 440/50 (nm)) and iRFP (excitation laser: 638 nm, emission: 720/30 (nm)). Individual clones highly expressing BFP or iRFP (top 2%) were respectively sorted into 96-well

tissue culture treated plates containing 200 uL fresh culture media. Clones were incubated for approximately 14 days until several clones reached sufficient population density. Populations were measured on the Attune NxT Flow Cytometer using the flow cytometry protocol to verify the expression of iRFP and BFP. Each clone was transfected with mCh and a mCh targeting guide, induced with GA or vehicle control, and run through flow cytometry to measure its mCh targeting knockdown ability. Clones that achieved robust knockdown in 2 transfection experiments in 2 weeks were maintained for future experiments.

## Lentivirus production and T cell transduction

N- and C-terminal Cas13 splits were cloned into 2 separate lentivirus vectors, and the gRNA expression cassette was cloned into the C-terminal Cas13 split virus vector. To produce lentivirus, lentivirus vectors, and viral packaging plasmids: pDelta, pAdv, and Vsvg were transfected to the LentiX cell line using Lipofectamine. After 6 h, culture media was replaced by Freestyle 293 Expression Medium (Fisher Scientific 12-338-026), supplemented with 50UI/ml penicillin/streptomycin (Corning 30001CI), L-glutamine (Corning 25005CI), 1 mM sodium pyruvate (Lonza 13115E) and 0.05 M sodium butyrate. The viral supernatant was collected twice at 24 h and 48 h after transfection. The collected viral supernatant was concentrated using the Lenti-X concentrator (Takara Bio 631232) following manufacturing protocol.

PBMCs were thawed two days before transduction and maintained in X-Vivo 15 media (Lonza 04-418Q) supplemented with 5% human AB serum (Valley Biomedical HP1022), 10mM N-acetyl L-Cysteine (Sigma A9165), 55 µM 2-Mercaptoethanol (Thermo Fisher 21985023), and 100 U/ml IL-2 (NCIBRB Preclinical Repository), with a density of 1,000,000 cells/ml. One day after thawing, PBMCs were activated with ImmunoCult™ Human CD3/CD28 T Cell Activator (STEMCELL 10971) according to manufacturing protocol. In the meantime, 12-well non-TC treated plates were coated with retronectin (Takara Bio T100B) at a concentration of 32 ng/ml in PBS. On the day of transduction (one day after T cell activation), retronectin-coated plates were incubated with 1% Bovine Serum Albumin (BSA) in PBS at room temperature for 30 min. Next, 1 ml of the concentrated virus was added to each well and then spun at 1200 g for 90 min. After the supernatant was removed, wells were washed with 1% BSA in PBS. 1 mL of activated PBMCs at a density of 250,000 cells/ml was added to virus-coated wells and spun down at 1200 × g for 10 min with the brake low. Cells were incubated overnight and transduced with virus 2 on the next day. Transduction was verified 3 days after using flow cytometry.

## Flow cytometry

For split site screening experiments, HEK293FT cells were resuspended for flow cytometry analysis 48 h post-induction. Cells were trypsinized using 0.05% trypsin/0.53 mM EDTA (Corning) and neutralized with 4x volume of 5PSGN. All cells were sampled in a Thermo Fisher Attune Nxt cytometer (a minimum of 10,000 transfected cells/sample). mCh fluorescence was detected by a 561 nm yellow laser and a 620/15 (nm) emission filter. iRFP720 expression was detected by using the 638 nm red laser and 720/30 (nm) bandpass emission filter. GFP was detected using a 488 nm blue laser and a 510/10 (nm) bandpass emission filter. BFP was detected using a 405 nm violet laser and 440/50 (nm) bandpass emission filter. For B2M knockdown experiments, HEK293FT cells were collected and surface-stained with either an HLA-ABC antibody conjugated with FITC (BD Bioscience 555552) or an HLA-A2 antibody conjugated by PE (BD Bioscience 560964). FITC signal was detected by a 488 nm blue laser and a 530/30 (nm) bandpass emission filter. PE signal was detected by a 561 nm yellow laser and a 585/16 (nm) bandpass emission filter. For CD46 knockdown in primary PBMC, cells were stained with Human CD46 Alexa Fluor 594-conjugated Antibody (R&D systems FAB2005T-1). Alexa Fluor 594 signal was detected with a 561 nm yellow laser and a 585/16 (nm) bandpass emission filter. Flow

cytometry data was analyzed using FlowJo (Treestar Software). Live cells were gated by forward scatter and side scatter. Transfected cells were gated for the presence of the iRFP transfection marker. Mean fluorescence intensities were calculated by FlowJo.

## Fluorescence imaging

Fluorescence microscopy was performed with Cytation 5 Cell Imaging Multimode Reader (BioTek) equipped with a BioSpa 8 Automated Microplate Incubator (BioTek). A 20× objective was used to collect bright-field images (autofocusing auto exposure), iRFP fluorescence images (623 LED; CY5 excitation 628/40 emission 685/40 filter cube), GFP fluorescence image (465 LED; GFP excitation 469/35, emission 525/39 filter cube), and mCh fluorescence image (554 LED; TRITC excitation 556/35, emission 600/39 filter cube). Cyation5 and BioSpa Automated Microplate Incubator (BioTek) were programmed to load plates for imaging every 2 h of incubation at 37 °C, 5% $CO_2$, and ~90% humidity. Image analysis was performed with Gen5 (BioTek).

## In vivo experiments

Plasmid DNA used for in vivo injections was extracted using EndoFree Plasmid Maxi Kit (Qiagen 12362) and dissolved in an endotoxin-free TE buffer at 1000 ng/ul. Female BALB/c mice (JAX 00651) 5–6 weeks old were transfected with plasmids carrying expression cassette for each split moiety, guide RNA, and target luciferase using hydrodynamic tail vein injection with TransIT-EE Delivery Solution (Mirus Bio MIR 5340). Mice were under anesthesia during injections. Transfected mice received intraperitoneal (i.p.) injections (100 mg/kg ABA, 10 mg/kg GA) of inducers 2 h and 6 h after hydrodynamic tail vein injections. Luminescence in mice liver was measured at 8 h (single luciferase) or 8 h and 24 h (dual-luciferase) after transfection by IVIS Spectrum (Xenogen) and was quantified as total flux (photons per sec) in the region of interest. Images were acquired within 10 min following i.p. injection of 150 mg/kg of D-luciferin (PerkinElmer 122799) or 10 min following i.p. injection of 28.75nmole of Fluorofurimazine (FFz) (Promega CS320501) in 200 ul PBS.

## Statistical methods

All transient gene expression experiments involved transfection of plasmid DNA into $n = 3$ separate cell cultures. Fluorescence intensities for each cell culture population were averaged and the standard error of mean or standard deviation was taken as noted. Data between two groups were compared using an unpaired two-tailed $t$-test. All curve fitting was performed with Prism 8 (GraphPad) and $p$-values are reported (not significant = $p > 0.05$, $*p < 0.05$, $**p < 0.01$, $***p < 0.001$). All error bars represented SEM.

## Inclusion and ethics

We work to ensure diversity in experimental samples by selecting from genomic datasets. Cas13 sequences from multiple organisms were considered in our designs, including sequences from diverse global origins. The final selection of the utilized components was made based on activity against several representative targets.

## Reporting summary

Further information on research design is available in the Nature Portfolio Reporting Summary linked to this article.

## Data availability

The plasmids are available upon request, but we may require payment and/or a completed Materials Transfer Agreement if there is potential for commercial application. Further information and requests for resources and reagents should be directed to and will be fulfilled by the lead contact, Wilson Wong (wilwong@bu.edu). Source data are provided with this paper.

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

## Acknowledgements

W.W.W. acknowledges funding from NIH Awards (U01CA265713, R01EB029483, R01EB031904, R01GM129011), NSF Awards 2027045. C.T. acknowledges funding from NIH R01GM129011-S1. We also thank Wong lab members for suggestions on the manuscript; Dr. Todd Blute from the BU Proteomics & Imaging Core Facility for flow cytometry assistance; BU IVIS imaging core facility for mice in vivo imaging (NIH grant 1S10RRO24523-01).

## Author contributions

Y.D. designed and generated genetic constructs, performed experiments, analyzed the data, generated figures, and drafted and edited the manuscript. C.T. designed and generated 4OHT-inducible constructs, performed experiments, analyzed the data, and edited the manuscript. J.H.C. performed RTqPCR experiments and experiments in neuro2a and Jurkat cell lines and analyzed the data. J.C. repeated screening experiments and analyzed the data. W.W.W. conceived and supervised the project and analyzed the data. All authors commented on and approved the paper.

## Competing interests

A patent application (application number: US18/066,482) has been filed by the Trustees of Boston University based on this work (inventors: Y.D. and W.W.W.). W.W.W. is a scientific co-founder and shareholder of Senti Biosciences and 4Immune. The remaining authors declare no competing interests.
