## [Peer Review File · Nature Communications]

Reviewers' Comments:

Reviewer #1:

Remarks to the Author:

In this manuscript Ding et al. design, develop and optimize a robust system to temporally control Cas13 activity using small molecules in different cell types ex vivo and in vivo. The system is orthogonal since can be applied to multiple Cas13 effectors and this allows multiplexed targeting that at the same time can be used to generate synthetic feedback circuits. The work increases the applications of the CRISPR-Cas13 system and opens up new approaches based in this RNA targeting technology. The manuscript is well written, figures are clear, and the analysis is solid.

However, I have two main questions that need to be addressed:

1) CRISPR-Cas13 systems have shown to induce collateral activity in mammalian cells. This approach could help to mitigate such as collateral activity since one could turn on and off Cas13 activity. However, in this work there is no any reference to the collateral activity (in some of the cell lines used have been described such as 293T cells). Could the authors check for the collateral activity in the experiments showed in the manuscript. At least, I believe that it would be interesting and necessary to see whether in conditions where collateral activity is clear the inducible systems described here could avoid it or decrease it. For example, a comparison using reporters can easily help to check this

2) The strategy to decrease the leakiness using NES and NLS domains in different parts of the proteins is nice but I do not quite understand how the system works. After small molecule induction how peptides isolated in separated compartments can meet each other? How a protein that is in nucleus by having NLS can go to cytosol after small molecule induction and vice versa how a peptide located in cytosol by NES can get imported to the nucleus? Indeed, do the authors know where RNA targeting is actually happening?

Reviewer #2:

Remarks to the Author:

The manuscript "Orthogonal inducible control of Cas13 circuits enables programmable RNA regulation in mammalian cells" by Yage Ding et al. presents CRISTAL, a platform for controlling RNA levels in mammalian cells, in cell cultures and in mice, with a set of engineered Cas13 proteins. The activity of these split Cas13 is triggered by fusion domains that dimerize in response to certain small molecules or endogenous signals. Furthermore, these enzymes could be combined to form complex genetic circuits.

The authors clearly describe the screening process and how they rationally approached the development of the CRISTAL platform. The findings are supported by an extensive amount of experimental data. In addition to the detailed characterization in vitro, the systems were also tested in mice to show the platform's applicability in vivo. Overall, this manuscript describes interesting results that merit publication in this journal.

Major remarks

-It might be relevant to give more information about the cleavage activities of the different Cas13 effectors. Cas13a for example exhibits collateral sequence non-specific cleavage activity. Thus, for the experiments in living cells, its impact on other RNAs might be relevant.

Minor remarks

- In the introduction, other applications of Cas13 systems could be mentioned, such for example the detection of viral RNA or miRNAs with Cas13a.

- There are some moments of ambiguity in the manuscript that require clarification: It is irritating reading "Cas13s" because it is not clear if it designates the split Cas13 variants or just the plural of Cas13. The terms "systematically outperformed" and "close to ideal" (Page 6, Lines 7-8) might be a little exaggerated as the trend seems not so clear when looking at the Supplementary Table 1 and the distance d is > 0.4 for most of them. Furthermore, the word "both" could be added (Page 15, Line 23), to make it more clear that not either of the two targeting RNAs is meant. In the experimental section, the emission wavelength for BFP and iRFP FACS is not stated (Page 36) and

the unit (nm) is lacking for the specifications of the bandpass emission filters and cube filters (Page 38, 39).

- The following points could be clarified in the figures and corresponding legends: Figure 1C: MCh and iRFP should be changed to the genes encoding for mCh and iRFP. Figure 1E: It is not clear what is meant by "across different sites". Figure 3A: It should be stated which split sides were used here. Figure 6B, C: It is not clear what exactly was compared ("and/or"). Figure 4B, C, Supplementary Figure C, D: It might be better not to represent genes like light bulbs. Supplementary Figure 1C: The distance d must be defined in the figure legend. Supplementary Table 1, 3: Please indicate what colored background and colored frames highlight. Supplementary Figure 5A: The title of the y-axis is might require a short explanation in the figure legend. Supplementary Figure 5G: The numeration from Day0 to Day4- should be detailed in the figure legend. The authors should consider reformulating the second sentence in the figure legend for more clarity. Supplementary Figure 7B: The authors should replace "6/8" with "Six out of eight". Supplementary Figure 15A, B: The empty fields should be labeled with "-GA" and "-Dano". Supplementary Figure 9B: The authors might replace the term "ERT2-C -NLS" with "ERT2-C-ERT2-NLS" (for N565-NLS/ERT2-566C-ERT2-NLS) for better clarity. The same applies to "N ERT2 -C-ERT2" instead of "N/ERT2 -C" (Page 9, Supplementary Figure 7B) and "ERT2-N-ERT2/C" instead of "N-ERT2/C" (Supplementary Figure 8B). In addition to that, the authors should rewrite the second sentence in the legend of Supplementary Figure 14A for better understandability. In some figures, the information about the number of replicates, error bars (Figures 1E, 2A, 3A, B, C, 5B, and Supplementary Figures), and colored areas (Figure 5B, Supplementary Figure 16B) is lacking. There are also some inconsistencies between the main text and some figures (Page 4: 27 split sites / Figure 1D: 28 arrows; Page 5: 10 split sites with inducibility >0.4 / Figure legend 1D: 8 sites).

In some figures (e.g. Figures 2C, 3A, Supplementary Figures 1A, 9C, D, 8B, 15A, B), the contrast has to be improved or contours (like in Supplementary Figure 4A) have to be added for better visibility.

- Furthermore, here some other possible changes will be proposed: The authors might specify the IC50 in nM instead of μM (Supplementary Figure 16D) and they should verify if the information in lines 5-7 on page 10 is correct (Supplementary Figure 9E: The inducibility of N-NES/NLS-C-NLS is decreased compared to N-NLS/C-NLS). In addition to that, the term N49-GAI/GID-50C in the legend of Supplementary Figure 13C should be replaced by N49-GID/GAI-50C, and the word "constitutively" is unsuitable in this context (Supplementary Figure 15D) as HLA expression depends on the absence or presence of Dano. In the last line of page 14, the reference should only be Supplementary Figure 16C, because Supplementary Figure 16A does not show any information about induction time and adaptation. For a better visibility, the font size of the axis label should be increased in Supplementary Figure 5G. In addition to that, it could be considered changing the order of the images so that subfigures D, E, and F are arranged one below the other (Supplementary Figure 5). Furthermore, there are several comments on Supplementary Figure 8C. The authors might arrange the elements in the table in the order of increasing split site number and the corresponding plot would be easier to read when the pairs with the same split sides were labelled with the same number. For the tables showing the distances to ideal performance, it could be stated that the formula shown in Supplementary Figure 1C was used to determine these values. Finally, the authors might think about removing some unnecessary cross-references to improve the readability of the text and Supplementary Tables 4 -6 as they are not referred to in the text.

- Some abbreviations should be introduced (CID (Page 5); i.v. and i.p. (Figure 6A); iRFP; NTgRNA, and tgRNA, KD (Supplementary Figure 2), NT (Supplementary Figure 10), WNT (Page 10)), others should be used throughout the document (mCh and Dano (Figure 5A, Supplementary Figures), GA instead of GIB (Page 7, 15 Figure 6B, Supplementary Figure 12B, 15C)). The authors might also consider using the abbreviations for the different ligands in Supplementary Table 2. Also, the term NS31a (Supplementary Figure 14E) should be introduced.

For better consistency, the same spelling (off/OFF, on/ON, μ / μ , with/without - (e.g. -inducible), h / hrs/hours, ml / mL, GIB / GA (e.g. Supplementary Figure 13E), -gRNA -> NTgRNA (Supplementary Figure 16D)), italics for certain terms (in vitro (e.g. Page 16), Escherichia coli (Page 32)), and font (for $^{\circ}\text{C}$) should be used.

Throughout the text, there were several typing errors (Figure 1C: fluorescence, Page 13 Line 17: 15A; Page 33: ., , Page 35: Transfected cells, Supplementary Figures 3, 9B, D, H, 13E, 14C, F, H: knockdown down, Page 7: EC50, Supplementary Figure 5C: PiggyBac, Supplementary Figure 9F: N507-N507-, Supplementary Figure 14H: prey, Page 12: Cas12). The authors might consider having the manuscript proofread.

Reviewer #3:

Remarks to the Author:

The authors engineered multiple split Cas13s and used chemically induced dimerization to control their activity. Their thorough optimization efforts led to good performances. The synbio and in vivo demonstrations are compelling. This paper serves as a strong starting point for further optimizing and validating these Cas13s for controlling endogenous transcripts.

My only request regards the known cytotoxicity of some Cas13s and their dependence on the cellular context. Although cytotoxicity info could be inferred from some of the controls in the paper, I'd appreciate it if the authors could provide it directly. Are the split Cas13s less toxic than the wild type? As toxic? Either way it is good to know. Related to that, is it possible that splitting reduces substrate specificity? Although asking for transcriptomic profiling wouldn't be reasonable for this revision, if the split versions prove more toxic, it might imply altered specificity and that would be good to know as well for the benefit of future users of these tools.

Response to Reviewers' Comments

We extend our sincere gratitude to all reviewers for their valuable and constructive feedback. Their positive assessment of our work is encouraging. It's worth highlighting that all three reviewers have emphasized the significance of addressing Cas13-related collateral activity, specifically in terms of comparing inducible split systems with their wild-type (WT) counterparts, prior to publication.

Since the inception of the RNA-targeting CRISPR-Cas13 system, collateral RNA cleavage stemming from on-target RNA binding and cleavage activity has been recognized *in vitro* and in bacterial contexts¹⁻⁴. While initial characterizations indicated minimal collateral activity in eukaryotic cells, Cas13-linked collateral activity has evolved into a pertinent concern within this system⁵⁻⁸. This phenomenon has been meticulously investigated across a spectrum of model systems and conditions, yielding some contradictory results attributed to variations in cell types, target transcripts, Cas13 orthologs, and gRNA choices⁶⁻⁹. However, there are discernible patterns that have emerged as common threads across studies, including the following key observations:

1. Collateral activity becomes more pronounced when Cas13 targets highly expressed transcripts^{7,8}.
2. Exogenous gene transcripts also display heightened susceptibility to collateral activity⁷⁻⁹.

Our investigations have verified the consistency of these principles within our studies, with an intensified emphasis on the comparison between inducible split systems and their WT counterparts. Through a series of novel experiments, we have effectively showcased the following outcomes:

1. Split Cas13 systems exhibit comparable or even improved collateral activity when juxtaposed with WT effector systems across diverse cell types. This observation extends to U87 cells, where significant Cas13 collateral activity has been noted^{5,9}.
2. A direct head-to-head evaluation of the two currently employed Cas13 orthologs in the CRISTAL platform, PspCas13b and RfxCas13d, has revealed that the former system yields lower collateral activity compared to the latter.
3. We have verified that the collateral activity of inducible Cas13 systems can be abated by opting for lower target expression levels, echoing the findings established by others.
4. Notably, our research has underscored the absence of cytotoxicity linked to collateral activity induced by both WT and split Cas13 systems across different cell lines.

In conclusion, we concur that the validation of Cas13-associated collateral activity is essential in new application contexts. Nevertheless, it's important to note that collateral activity, along with any potential associated toxicity, can be effectively mitigated through careful spacer sequence design, Cas13 ortholog selection, the choice of targets with low expression levels, and tunable Cas13 activity as offered by our CRISTAL platform¹⁰. Moreover, the inducible split Cas13 systems elucidated within this manuscript demonstrate comparable or even improved specificity compared to the WT.

Considering these findings, we are confident that our work not only contributes to the comprehensive understanding of Cas13-related collateral activity but also provides insights into harnessing the inducible split Cas13 systems for enhanced specificity and controllable activity.

Within the manuscript, text discussing collateral activity have been included in the manuscript starting Line 20 Page 17. Data is incorporated in Supplementary Figure 17 and 18. Any modifications or additions to the manuscript are shown as track changes. Our point-by-point responses to reviewer remarks are indicated in blue text in this document.

Reviewer #1 (Remarks to the Author):

In this manuscript Ding et al. design, develop and optimize a robust system to temporally control Cas13 activity using small molecules in different cell types ex vivo and in vivo. The system is orthogonal since can be applied to multiple Cas13 effectors and this allows multiplexed targeting that at the same time can be used to generate synthetic feedback circuits. The work increases the applications of the CRISPR-Cas13 system and opens up new approaches based in this RNA targeting technology. The manuscript is well written, figures are clear, and the analysis is solid.

Thank you for the positive comments.

However, I have two main questions that need to be addressed:

1) CRISPR-Cas13 systems have shown to induce collateral activity in mammalian cells. This approach could help to mitigate such as collateral activity since one could turn on and off Cas13 activity. However, in this work there is no any reference to the collateral activity (in some of the cell lines used have been described such as 293T cells). Could the authors check for the collateral activity in the experiments showed in the manuscript. At least, I believe that it would be interesting and necessary to see whether in conditions where collateral activity is clear the inducible systems described here could avoid it or decrease it. For example, a comparison using reporters can easily help to check this

Response to “However, in this work there is no any reference to the collateral activity (in some of the cell lines used have been described such as 293T cells). Could the authors check for the collateral activity in the experiments showed in the manuscript.”:

Thank you for your recognition of our work in demonstrating collateral activity in HEK cells. In our screening experiments, we took iRFP expression into account to address variations in transfection efficiency and the knockdown specificity of the systems being tested. We've depicted the expression levels of iRFP through flow cytometry dot plots, showcased in Supplementary Figures 3, 9, 13, and 14. While mCh knockdown is evident in the "on" state of the tested systems, as indicated by the downward shift of the cell population in the dot plots, the expression of iRFP remains relatively unchanged. This observation suggests minimal collateral activity against iRFP, induced by inducible split Cas13-mediated mCh-targeting knockdown.

Beyond HEK cells, we extended the application of CRISTAL systems to Neuro2A cells, Jurkat cells, and primary PBMCs. Since we consistently employed iRFP as the bystander gene, we analyzed iRFP expression in the presence and absence of mCh-targeting knockdown activity.

In Neuro2A cells, there is no significant change in iRFP expression between non-target and mCh targeting conditions for WT PspCas13b, GA-induced split Cas13b, and GA-induced split Cas13d (Figure IA). Despite these effector systems displaying highly efficient (>0.8) on-target mCh knockdown activity, minimal changes in iRFP expression suggest limited collateral activity against iRFP under these conditions (Figure IA). While on-target activity of neither WT PspCas13b nor the GA-induced split Cas13b demonstrates collateral activity against iRFP, WT RfxCas13d notably decreased iRFP expression with an mCh-targeting gRNA. This effect wasn't observed with the GA-induced split RfxCas13d, despite comparable on-target activity to the WT. This suggests that the GA-induced split RfxCas13d maintains WT-comparable on-target activity while minimizing collateral activity against iRFP in Neuro2A cells (Figure IA).

In Jurkat cells, both WT RfxCas13d and GA-induced split RfxCas13d led to a significant reduction (~20%) in iRFP expression when targeting mCh with similar efficiency, suggesting comparable collateral activity by these systems in Jurkat cells (Figure IB).

For primary PBMCs, GA-induced split RfxCas13 with either an NT gRNA or a CD46-targeting gRNA were introduced via lentivirus transduction. One Cas13 split fragment was expressed along with iRFP to enable the selection of successfully transduced PBMC populations. Following GA induction, the GA-induced split RfxCas13d induced CD46 knockdown in approximately 33% of transduced PBMCs. Notably, the expression of iRFP remained unchanged, suggesting minimal collateral activity generated by the CD-46 targeting activity of the GA-induced split Cas13d system in PBMCs (Figure IC).

Response to “At least, I believe that it would be interesting and necessary to see whether in conditions where collateral activity is clear the inducible systems described here could avoid it or decrease it.”

We strongly agree that collateral activity should be tested in a condition where it is known to be clear. According to the literature, various Cas13 orthologs show significant collateral activity in U87 cells, to a point where significant cytotoxicity is observed for Cas13^{5,9}. In addition, GFP expression have been found to be susceptible to mCh-targeting Cas13 activity induced collateral activity^{5,7,8}. Therefore, we further characterized Cas13 collateral activity against GFP in U87 cells as an experiment system with clearer collateral activity (Figure II).

With this new experimental design, WT PspCas13b with an mCh-targeting guide generated significant collateral knockdown of GFP (~60%) with strong on-target activity knocking down mCh expression (>80%) (Figure IIA, B). Notably, Dano-inhibited split PspCas13b generated minimal collateral activity against GFP while knocking down target expression by ~40% with the mCh-targeting gRNA (Figure IIA, D). Something to note here is the moderate on-target knockdown of mCh by Dano-inhibited split PspCas13b is hypothesized to contributed to the reduced collateral activity. Regarding the reduced on-target activity of Dano-inhibited split PspCas13b in U87 cells compared to in HEK cells, it is possible that the limited transfection efficiency in U87 cells resulted in limited co-transfection of both split fragments into the same cells, and on-target activity may be improved with higher transfection efficiency in U87 cells. In addition, we also observed that the substantial collateral activity by WT PspCas13b against GFP reduces with a 10-fold decrease in the amount of target (mCh) encoding plasmid transfected, while Dano-inhibited split Cas13b generated minimal collateral knockdown of GFP with consistent on-target knockdown efficiency against mCh at different expression level (Figure IIB, D).

Parallel to the behavior of WT PspCas13b, WT RfxCas13d also triggers considerable collateral activity against GFP when targeting mCh (Figure IIIA). This collateral knockdown of GFP diminishes from approximately 60% to 40% with a 10-fold reduction in the transfected mCh-encoding plasmid (Figure IIIA). However, the split systems yield varying results: split PspCas13b exhibits minimal collateral activity, while split RfxCas13d systems do not (Figure IIIB-E). Although split RfxCas13d systems exhibits on-target and collateral activity of comparable efficiency to the WT, their collateral activity is regulated by the inducer concentration along with their on-target activity (Figure IIID, E). Moreover, when target expression is lowered, the collateral activity against GFP by the GA-inducible split RfxCas13d notably drops from around 60% to 40%, with no adverse impact on the on-target mCh activity (~80%) (Figure IIIA, C, and E). Moreover, although this trend is less apparent in the Dano-inhibited split Cas13d system.

In conclusion, our work in U87 cells underscores that when dealing with highly expressed transgene knockdown, the WT RfxCas13d, split RfxCas13d, and WT PspCas13b exhibit significant collateral activity against a transgene reporter. Conversely, the Dano-inhibited split PspCas13b system demonstrates minimal collateral activity, offering a promising choice for application in U87 cells. For researchers considering the use of other WT or split Cas13 systems in U87 cells, pursuing reduced "on" activity and opting for lower target expression levels seems to be a productive strategy to mitigate collateral activity effects.

Figure I. Cas13 facilitated mCh-targeting knockdown in Neuro2A, Jurkat, and primary PBMC did not show collateral knockdown against iRFP expression.

A. In Neuro2A cells, WT PspCas13b and GA-inducible split Cas13b do not reduced iRFP expression when knocking down mCh. iRFP expression maintains consistent between no gRNA groups and mCh-gRNA groups for WT PspCas13b and GA induced GA-inducible split Cas13b. WT RfxCas13d, on the other hand, causes significant iRFP expression reduction with mCh knockdown, resulting in a collateral activity of 0.21 (collateral activity = (relative iRFP_{no gRNA} - relative iRFP_{gRNA}) / relative iRFP_{no gRNA}, relative iRFP = iRFP MFI_{testing} / iRFP_{iRFP mCh only}). GA-inducible split Cas13d generates 0.13 collateral activity against iRFP with GA induced mCh-targeting knockdown.

- B. In Jurkat cells, WT RfxCas13d, causes significant iRFP expression reduction with mCh knockdown, resulting in a collateral activity of 0.26. GA-inducible split Cas13d generates 0.21 collateral activity against iRFP with GA induced mCh knockdown.
- C. In PBMC, GA induces knockdown of CD46 in 33% transduced cells, while having no significant impact on iRFP expression in the same cell population.

Figure II. WT PspCas13b mediated mCh knockdown induces strong collateral activity against GFP in U87 cells, while Dano-inhibited split Cas13b demonstrates minimal bystander GFP knockdown.

- A. WT PspCas13b knocks down GFP when it is targeted to knockdown mCh with an mCh target gRNA. Dano-inhibited split Cas13b does not affect bystander GFP expression when knocking down mCh specifically in absence of Dano.
- B. Target (mCh) expression induces collateral activity of WT PspCas13b against GFP in a dose dependent manner. With a high dose of mCh plasmid transfected, GFP knockdown is as high as 60% while on target mCh knockdown is ~90%. With 0.2x10ng mCh plasmid transfected, collateral knockdown of GFP is reduced to ~40%, while on-target mCh knockdown efficiency remains to be ~90%. With no mCh expression, GFP is not targeted by WT PspCas13b and its mCh-targeting gRNA. $KD\ efficacy\ (\%) = (mCh\ or\ GFP\ MFI_{no\ gRNA} - mCh\ or\ GFP\ MFI_{gRNA}) / mCh\ or\ GFP\ MFI_{no\ gRNA} \times 100\%$
- C. Dano-inhibited split Cas13b generates Dano dose dependent knockdown of mCh with minimal impact on GFP expression.

D. Dano-inhibited split Cas13b shows minimal collateral activity against GFP with various mCh (target) plasmid transfected.

Figure III. Split RfxCas13d shows comparable level of collateral activity against GFP and on-target activity against mCh in U87 cells.

A. Collateral activity of WT RfxCas13d against GFP is dependent on mCh expression level. With a high dose of mCh plasmid transfected, GFP knockdown is as high as 60% while on target mCh knockdown is ~90%. With 0.2x10ng mCh plasmid transfected, collateral knockdown of GFP is reduced to ~40% while on-target mCh knockdown efficiency remains to be ~90%. With no mCh expression, GFP is not targeted by the mCh-targeting gRNA. $KD\ efficacy\ (\%) = \frac{(mCh\ or\ GFP\ MFI_{no\ gRNA} - mCh\ or\ GFP\ MFI_{gRNA})}{mCh\ or\ GFP\ MFI_{no\ gRNA}} \times 100\%$

B. Dano-inhibited split RfxCas13d knocks down target (mCh) and bystander (GFP) with similar KD efficacy with high or low mCh expression conditions.

C. GA-inducible split RfxCas13d shows reduced collateral activity against GFP with reduced target expression level (reduced amount of mCh-encoding plasmid transfected).

D. Dano-inhibited split RfxCas13d knocks down mCh and GFP in a Dano dose-dependent manner: both collateral and on-target activity reduce with higher dose of Dano as demonstrated

by the increasing expression of GFP and mCh. Reducing target dose does not significantly affect the on-target activity and collateral activity. MFI of GFP and mCh are normalized to the no gRNA no inducer control transfected with the same target and effector plasmid dose.

E. GA-inducible split RfxCas13d knocks down mCh and GFP in a GA dose-dependent manner: both collateral and on-target activity increase with higher dose of GA as demonstrated by the decreasing expression of GFP and mCh. Reducing target dose minimally affects on-target activity while decreases collateral activity against GFP. MFI of GFP and mCh are normalized to the no gRNA no inducer control transfected with the same target and effector plasmid dose.

2) The strategy to decrease the leakiness using NES and NLS domains in different parts of the proteins is nice but I do not quite understand how the system works. After small molecule induction how peptides isolated in separated compartments can meet each other? How a protein that is in nucleus by having NLS can go to cytosol after small molecule induction and vice versa how a peptide located in cytosol by NES can get imported to the nucleus? Indeed, do the authors know where RNA targeting is actually happening?

Response:

Thank you for your insightful comment. We have two theories to explain the mechanism underlying the induced dimerization of the nuclear localization signal (NLS) and nuclear export signal (NES) tagged Cas13 splits:

1. In the context of cell division, the nuclear membrane undergoes a breakdown. During this process, split Cas13 moieties that were previously sequestered within the nucleus (with NLS tags) will be released. These split moieties could then bind to their complementary counterparts within the cells when the inducer is present. After cell division, the dimerized split Cas13 systems will persist in their dimerized state and remains to be active in whichever cellular compartments they are located, cytosol or nuclei.
2. Another possibility involves the synthesis of new split Cas13 proteins in the cytosol. When a split Cas13 is tagged with an NLS is synthesized, it remains in the cytosol until its NLS is recognized by the nuclear membrane pore's importing machinery. Consequently, the NLS-tagged Cas13 split might have a chance to encounter its complementary split in the cytosol before committing to nuclear transport.

Additionally, it's important to note that the shuttling of proteins facilitated by NLS and NES across the nuclear membrane is not a permanent process. The dynamic distribution of NLS-tagged proteins tends to favor the nucleus, while NES-tagged proteins are skewed towards the cytosol. This leads to a lower abundance of NLS-tagged proteins in the cytosol and vice versa, but these NLS-tags proteins in the cytosol should still be able to bind its complement in the cytosol upon induction and become active.

Reviewer #2 (Remarks to the Author):

The manuscript "Orthogonal inducible control of Cas13 circuits enables programmable RNA regulation in mammalian cells" by Yage Ding et al. presents CRISTAL, a platform for controlling RNA levels in mammalian cells, in cell cultures and in mice, with a set of engineered Cas13 proteins. The activity of these split Cas13 is triggered by fusion domains that dimerize in response to certain small molecules or endogenous signals. Furthermore, these enzymes could be combined to form complex genetic circuits.

The authors clearly describe the screening process and how they rationally approached the development of the CRISTAL platform. The findings are supported by an extensive amount of

experimental data. In addition to the detailed characterization in vitro, the systems were also tested in mice to show the platform's applicability in vivo. Overall, this manuscript describes interesting results that merit publication in this journal.

Major remarks

-It might be relevant to give more information about the cleavage activities of the different Cas13 effectors. Cas13a for example exhibits collateral sequence non-specific cleavage activity. Thus, for the experiments in living cells, its impact on other RNAs might be relevant.

Response:

Thank you for your positive feedback. To assess the cleavage efficiencies of various Cas13 effectors, we opted to focus on PspCas13b, RfxCas13d, and their corresponding CRISTAL systems for characterization. Cas13a is not included in our analysis due to its less effective knockdown performance in both its wild-type and split forms during our experiments.

We use a transgene reporter system in HEK cells for the direct and quantitative comparison of collateral activity between the 2 Cas13 orthologs (Figure IV). Our experimental setup involves transfecting HEK cells with plasmids encoding the respective Cas13 effector systems, their corresponding mCh-targeting gRNA, or a blank vector for the non-targeted condition, along with mCh (target), GFP (collateral activity reporter), and iRFP (transfection marker). By employing fluorescence imaging, we measure the mean fluorescence intensity (MFI) of mCh and GFP within the transfected (iRFP positive) HEK cell population.

Regarding the PspCas13b effector systems, we note that on-target activity against mCh by either the WT effector or the Dano-inhibited split PspCas13b in the absence of Dano does not lead to a significant reduction in GFP expression, suggesting minimal collateral activity of these 2 systems in HEK cells (Figure IVA, B). Furthermore, this minimal collateral activity against GFP remains consistent even over a longer duration of Cas13 expression and activity (72 hours compared to the initial 48 hours post-transfection) (Figure IVA, B). Expanding beyond the transgene reporter system depicted in Figure II, it's important to highlight that the on-target activity against mCh exhibited by the Dano-inhibited split PspCas13b system does not exert an impact on endogenous gene expression (B2M) in HEK cells (Figure IVC, D)

In the context of the RfxCas13d effector systems, we observed collateral knockdown of GFP expression as efficient as targeted mCh knockdown when utilizing the WT Cas13d, specifically within 48 hours post-transfection (Figure IVC). While collateral knockdown of GFP expression by the Dano-inhibited split Cas13d system is not as substantial as that by the WT Cas13d, the downregulation of GFP expression is still significant in the absence of Dano (Figure IVC). Notably, a divergence in collateral activity towards transgene expression between Cas13b and Cas13d has also been observed in other eukaryotic cell lines, such as Drosophila cells and HeLa cells⁸.

Figure IV. WT and Dano-inhibited split Cas13b shows minimal collateral activity against GFP when targeting mCh, while WT and Dano-inhibited RfxCas13d demonstrates great collateral activity against GFP when targeting mCh.

- A. 48h after transfected with the mCh-targeting Cas13b systems in HEK cells, GFP expression stays unaffected when mCh is specifically knocked down by the Dano-inhibited split Cas13b in absence of Dano and by WT PspCas13b.
- B. 72h after transfected with the mCh-targeting Cas13b systems in HEK cells, GFP expression stays unaffected when mCh is specifically knocked down by the Dano-inhibited split Cas13b in absence of Dano and by WT PspCas13b.
- C. 48h after transfected with the mCh-targeting Cas13d systems in HEK cells, GFP expression is reduced with mCh-targeting Dano-inhibited split Cas13d in absence of Dano and by WT RfxCas13d.
- D. 72h after transfected with the mCh-targeting Cas13d systems in HEK cells, GFP expression is reduced along with mCh with mCh-targeting by the Dano-inhibited split Cas13d in absence of Dano and by WT RfxCas13d.

Previous studies have demonstrated that the extent of RfxCas13d-associated collateral activity is influenced by factors such as the cell line used, the specific gRNA sequence employed, and the reporter system utilized^{8,9}. Building upon this knowledge, our aim is to establish well-defined experimental guidelines that facilitate the effective and streamlined utilization of inducible split RfxCas13d systems while minimizing collateral activity.

To achieve this objective, we examined collateral activity in HEK cells using the GFP collateral activity reporter system. This approach allows us to explore collateral activity across a range of effector activity levels, effector expression levels, and target expression levels (Figure V). With decreasing WT RfxCas13d encoding plasmid dosage and increasing Dano concentration, collateral activity against GFP decreases along with the on-target activity against mCh (Figure V). This suggests that attaining potent on-target activity while simultaneously minimizing collateral activity through Cas13 activity regulation might be a challenging endeavor.

Conversely, by modulating the level of target expression, we observed a notable decline in collateral activity with minimal impact on on-target activity. Collateral activity against GFP by both WT and Dano-inhibited split RfxCas13d decreases with reduced amount of mCh (target) encoding plasmid transfected (Figure VA, C). Minimal GFP knockdown is observed with the mCh-targeting gRNA in absence of mCh, suggesting GFP expression knockdown is induced by the on-target activity of Cas13d instead of off-target binding of the mCh-targeting gRNA to GFP sequence (Figure VA, C). More importantly, on-target activity is not reduced with the decreasing target expression level (Figure VB, D). These results suggest that lowering target expression level enhances specificity of RfxCas13d systems without negatively affecting their on-target activity.

Similar correlations between target expression level and RfxCas13 collateral activity has been established using transgene reporter systems in cell line such as *Drosophila* and HeLa cells⁸. Although substantial collateral activity against transgene by RfxCas13 seems to be consistent when targeting another highly expressed transgene, RfxCas13d targeting endogenous transcripts that have lower copy numbers and are expressed at a lower level is less likely to suffer from substantial collateral activity. And indeed, in our experiments where we applied split RfxCas13d system for endogenous gene knockdown demonstrated minimal collateral activity against untargeted endogenous transcript (Figure IIC) or transgene expression (Supplementary Figure 5D-F).

To sum up, the collateral activity against a transgene reporter triggered by RfxCas13d's on-target activity against a highly expressed transgene is substantial across various cell types and targets. However, this collateral activity can be attenuated through adjustments such as reducing the expression level of the target, lowering the on-activity of CRISTAL Cas13d systems, or moderating the expression level of RfxCas13d specifically in HEK cells. Furthermore, in our investigations, both the wild-type PspCas13b and the inducible split PspCas13b systems displayed minimal collateral activity when targeting both transgene and endogenous gene expressions in HEK cells, even when dealing with highly expressed targets. In a broader context, the successful application of the CRISPR-Cas13 system, along with the CRISTAL platform, hinges on the strategic fine-tuning of Cas13 effectors using our inducible systems. This, combined with a thoughtful selection of the most effective system for a given experimental context, is poised to yield positive outcomes.

Figure V. Collateral activity against GFP induced by mCh-targeting activity of WT or split RfxCas13d in HEK cells decreases with reduced Cas13 activity and target expression.

- A. Collateral activity against GFP induced by mCh-targeting activity of WT RfxCas13d decreases with reduced WT RfxCas13d expression level (reduced plasmid dose transfected) and reduced target expression (reduced mCh-encoding plasmids transfected) in HEK cells. Collateral activity = $(\text{GFP MFI}_{\text{no gRNA}} - \text{GFP MFI}_{\text{mCh gRNA}}) / \text{GFP MFI}_{\text{no gRNA}}$. On-target activity = $(\text{mCh MFI}_{\text{no gRNA}} - \text{mCh MFI}_{\text{mCh gRNA}}) / \text{mCh MFI}_{\text{no gRNA}}$.
- B. On-target activity WT RfxCas13d decreases mainly with reduced WT RfxCas13d expression level, while remaining relatively consistent with varied target expression level.
- C. Collateral activity against GFP induced by mCh-targeting activity of Dano-inhibited split RfxCas13d decreases with reduced Cas13d activity (increased Dano conc.) and reduced target expression (reduced mCh-encoding plasmids transfected) in HEK cells.
- D. On-target activity Dano-inhibited split RfxCas13d decreases mainly with reduced WT RfxCas13d expression level, while remaining relatively consistent with varied target expression level.

Minor remarks

- In the introduction, other applications of Cas13 systems could be mentioned, such for example the detection of viral RNA or miRNAs with Cas13a.

Thank you for the comments. We have included a reference to this important development in the introduction section.

There are some moments of ambiguity in the manuscript that require clarification: It is irritating reading “Cas13s” because it is not clear if it designates the split Cas13 variants or just the plural of Cas13.

Thank you for the comment. We have changed Cas13s (plural) to Cas13 effectors to signify plural of Cas13.

The terms “systematically outperformed” and “close to ideal” (Page 6, Lines 7-8) might be a little exaggerated as the trend seems not so clear when looking at the Supplementary Table 1 and the distance d is > 0.4 for most of them.

We have changed these terms “substantial improvement” to tone down the claims.

Furthermore, the word “both” could be added (Page 15, Line 23), to make it more clear that not either of the two targeting RNAs is meant.

We added the word “both” to the sentence.

In the experimental section, the emission wavelength for BFP and iRFP FACS is not stated (Page 36) and the unit (nm) is lacking for the specifications of the bandpass emission filters and cube filters (Page 38, 39).

Thank you for the note. We have addressed the mistakes.

- The following points could be clarified in the figures and corresponding legends: Figure 1C: MCh and iRFP should be changed to the genes encoding for mCh and iRFP.

Thank you. We have made this change.

Figure 1E: It is not clear what is meant by “across different sites”.

Thank you. We have changed it to “with more split sites”

Figure 3A: It should be stated which split sides were used here.

We have included this information in the illustration.

Figure 6B, C: It is not clear what exactly was compared (“and/or”).

We have clarified that the comparison is between induced versus uninduced targeted groups.

Figure 4B, C, Supplementary Figure C, D: It might be better not to represent genes like light bulbs.

We kept this illustration as it represents the gene being turned on and off.

Supplementary Figure 1C: The distance d must be defined in the figure legend.

We have made this addition to the legend.

Supplementary Table 1, 3: Please indicate what colored background and colored frames highlight.

We have updated these tables with legends.

Supplementary Figure 5A: The title of the y-axis might require a short explanation in the figure legend.

We have added an explanation.

Supplementary Figure 5G: The numeration from Day0 to Day4- should be detailed in the figure legend. The authors should consider reformulating the second sentence in the figure legend for more clarity.

We have added a reference to the method section “Stable cell line generation and reversibility experiment” and explained numeration in the legend.

Supplementary Figure 7B: The authors should replace “6/8” with “Six out of eight”.

We have made this change.

Supplementary Figure 15A, B: The empty fields should be labeled with “-GA” and “-Dano”.

We have made this edit.

Supplementary Figure 9B: The authors might replace the term “ERT2-C -NLS” with “ERT2-C-ERT2-NLS” (for N565-NLS/ERT2-566C-ERT2-NLS) for better clarity. The same applies to “N-ERT2 -C-ERT2” instead of “N/ERT2 -C” (Page 9, Supplementary Figure 7B) and “ERT2-N-ERT2/C” instead of “N-ERT2/C” (Supplementary Figure 8B).

Our original legend are correct with one ERT2 domain linked to the split Cas13 fragments in these conditions.

In addition to that, the authors should rewrite the second sentence in the legend of Supplementary Figure 14A for better understandability.

We edited the second sentence.

In some figures, the information about the number of replicates, error bars (Figures 1E, 2A, 3A, B, C, 5B, and Supplementary Figures), and colored areas (Figure 5B, Supplementary Figure 16B) is lacking.

There are also some inconsistencies between the main text and some figures (Page 4: 27 split sites / Figure 1D: 28 arrows; Page 5: 10 split sites with inducibility >0.4 / Figure legend 1D: 8 sites).

We have corrected these errors.

In some figures (e.g. Figures 2C, 3A, Supplementary Figures 1A, 9C, D, 8B, 15A, B), the contrast has to be improved or contours (like in Supplementary Figure 4A) have to be added for better visibility.- Furthermore, here some other possible changes will be proposed: The authors might specify the IC50 in nM instead of μM (Supplementary Figure 16D) and they should verify if the information in lines 5-7 on page 10 is correct (Supplementary Figure 9E: The inducibility of N-NES/NLS-C-NLS is decreased compared to N-NLS/C-NLS).

We have removed “Dano-inducible” from this statement since inducibility for this system is decreased with its leakiness upon optimization.

In addition to that, the term N49-GAI/GID-50C in the legend of Supplementary Figure 13C should be replaced by **N49-GID/GAI-50C**, and the word “**constitutively**” is unsuitable in this context (Supplementary Figure 15D) as HLA expression depends on the absence or presence of Dano.

We have made these changes.

In the last line of page 14, the reference should only be **Supplementary Figure 16C**, because Supplementary Figure 16A does not show any information about induction time and adaptation.

We have made this edit.

For a better visibility, the font size of the axis label should be increased in Supplementary Figure 5G.

We have made this edit.

In addition to that, it could be considered changing the order of the images so that subfigures D, E, and F are arranged one below the other (Supplementary Figure 5).

Furthermore, there are several comments on Supplementary Figure 8C. The authors might arrange the elements in the table in the order of increasing split site number and the corresponding plot would be easier to read when the pairs with the same split sides were labelled with the same number. For the tables showing the distances to ideal performance, it could be stated that the formula shown in Supplementary Figure 1C was used to determine these values. Finally, the authors might think about removing some unnecessary cross-references to improve the readability of the text and Supplementary Tables 4 -6 as they are not referred to in the text.

- Some abbreviations should be introduced (CID (Page 5); **i.v. and i.p.** (Figure 6A); iRFP; NTgRNA, and tgRNA, KD (Supplementary Figure 2), NT (Supplementary Figure 10), WNT (Page 10)), others should be used throughout the document (**mCh and Dano** (Figure 5A, Supplementary Figures), **GA instead of GIB** (Page 7, 15 Figure 6B, Supplementary Figure 12B, 15C)). The authors might also consider using the abbreviations for the different ligands in **Supplementary Table 2**. Also, the term NS31a (Supplementary Figure 14E) should be introduced.

Thank you for the note. We have made these edits.

For better consistency, the same spelling (**off/OFF, on/ON**, u / μ , **with/without – (e.g. - inducible)**, h / hrs/hours, **ml / mL**, GIB / GA (e.g. Supplementary Figure 13E), -gRNA -> NTgRNA (Supplementary Figure 16D)), italics for certain terms (**in vitro (e.g. Page 16)**, **Escherichia coli (Page 32)**), and font (for °C) should be used.

We have made these edits.

Throughout the text, there were several typing errors (Figure 1C: fluorescence, Page 13 Line 17: 15A; Page 33: ., , Page 35: **Transfected cells**, Supplementary Figures 3, 9B, D, H, 13E, 14C, F, H: knockdown down, Page 7: **EC50**, Supplementary Figure 5C: **PiggyBac**, Supplementary Figure 9F: N507-N507-, Supplementary Figure 14H: prey, Page 12: Cas12). The authors might consider having the manuscript proofread.

Thank you and we have made these edits.

Reviewer #3 (Remarks to the Author):

The authors engineered multiple split Cas13s and used chemically induced dimerization to control their activity. Their thorough optimization efforts led to good performances. The synbio and in vivo demonstrations are compelling. This paper serves as a strong starting point for further optimizing and validating these Cas13s for controlling endogenous transcripts.

Thank you for the positive feedback.

My only request regards the known cytotoxicity of some Cas13s and their dependence on the cellular context. Although cytotoxicity info could be inferred from some of the controls in the paper, I'd appreciate it if the authors could provide it directly. Are the split Cas13s less toxic than the wild type? As toxic? Either way it is good to know. Related to that, is it possible that splitting reduces substrate specificity? Although asking for transcriptomic profiling wouldn't be reasonable for this revision, if the split versions prove more toxic, it might imply altered specificity and that would be good to know as well for the benefit of future users of these tools.

Response:

Thank you for your comment. The Cas13 effectors, particularly RfxCas13d, have been reported to exhibit cytotoxic effects in different cell lines like Hela, U87, and HepG2 cells, as referenced. However, the specific mechanism behind the cytotoxicity associated with Cas13d and Cas13b remains unclear. Ai et al. investigated the activity of RfxCas13d targeting various luciferase genes in Hela cells. While the authors studied collateral activity by the WT RfxCas13d with various targeting gRNA sequences and against different untargeted luciferases, they used reduced RNA yield as the evidence of cytotoxicity, and no direct cytotoxicity assays were conducted in this study. It is also worth noting that reduced RNA yield seems to depend on the gRNA sequence. With one RLuc-targeting gRNA sequence, despite considerable collateral knockdown against nLuc, there was no decrease in RNA yield compared to the non-target control.

In the context of U87 cells, Ozcan et al. discovered that both PspCas13b and RfxCas13d led to decreased cell viability compared to their newly identified Cas7-11 effector system. Interestingly, this cytotoxicity does not appear to be associated with the activity of these systems: when comparing non-targeting conditions with GLuc-targeting conditions, there was no substantial difference in cell viability (no reduction for RfxCas13d and ~5% for PspCas13b), despite RfxCas13d displaying substantial non-specific activity in a whole transcriptome analysis. Therefore, according to this study, cytotoxicity is not completely associated with the collateral activity of WT RfxCas13d and PspCas13b in U87 cells.

When examining HepG2 cells, Ozcan et al. also found reduced cell viability by RfxCas13d mediated knockdown of GLuc expression. However, this cytotoxicity is not consistent across all gRNAs that effectively reduced GLuc expression.

To sum up, the cytotoxicity associated with RfxCas13d and PspCas13b seems to be influenced by factors such as the cell type (Ai et al. observed no reduction of RNA yield related to collateral activity of RfxCas13d in HEK cells) and the specific gRNA used. More importantly, not all cytotoxicity is related to collateral activity and not all collateral activity leads to cytotoxicity.

In our study, we compared the collateral activity against a transgene reporter (GFP) and the cytotoxicity related to the mCh-targeting activity of both the WT and split versions of Cas13 effector systems in HEK and U87 cells. Our findings indicate that neither the WT nor the split Cas13 systems show activity-related cytotoxicity. Specifically, when considering RfxCas13d, all tested effector systems demonstrate significant collateral activity in both HEK and U87 cells. However, there is not a substantial change in viability under mCh-targeting conditions compared to their non-targeted conditions (Figure VIA, C). The normalized viability remains close to 1 (Figure VIA, C). As for the PspCas13b system, neither the WT nor the split Cas13b versions demonstrated collateral activity or cytotoxicity in HEK cells (Figure VIB). In U87 cells, while WT PspCas13b exhibits notable collateral activity compared to the Dano-inhibited split PspCas13b, it does not result in reduced viability when targeting mCh (Figure VID).

In summary, the connection between Cas13 collateral activity and cytotoxicity remains elusive, as observed by us and others^{5,7-10}. Our own findings reveal that split Cas13b and split Cas13d exhibit comparable cytotoxicity to their wild-type counterparts in both HEK cells and U87 cells, while exhibiting distinct levels of collateral activity. Using a transgene reporter system, we identified that the Dano-inhibited split PspCas13b displayed minimal collateral activity in both HEK and U87 cells. Similarly, the WT PspCas13b exhibited minimal collateral activity in HEK cells. On the other hand, the RfxCas13d system consistently displayed significant collateral activity against GFP when targeting mCh expression across both cell lines.

In light of these findings, it's worth considering the potential of utilizing split PspCas13b systems for enhanced specificity. Additionally, while certain effectors exhibited notable collateral activity in the transgene reporter system, this activity demonstrated a correlation with the level of target expression and the degree of on-target activity. Consequently, an important design principle that we propose is to implement controlled effector activity against less abundant targets. This approach could effectively minimize both collateral activity and cytotoxicity.

Figure VI. Cytotoxicity and collateral activity of WT Cas13 effectors and split Cas13 effectors in HEK and U87 cells.

A. In HEK cells, Dano-inhibited and GA-inducible split RfxCas13d do not show significantly reduced collateral activity against GFP when targeting mCh compared to the WT RfxCas13d. Similarly, there is no significant difference among the viability change associated with mCh knockdown by the 3 Cas13d systems. Collateral activity = $(\text{GFP MFI}_{\text{no gRNA}} - \text{GFP MFI}_{\text{mCh gRNA}}) / \text{GFP MFI}_{\text{no gRNA}}$. Normalized viability = $\text{Luminescence}_{\text{no gRNA}} / \text{Luminescence}_{\text{mCh gRNA}}$.

B. In HEK cells, neither Dano-inhibited split PspCas13b and WT PspCas13b show collateral knockdown of GFP when targeting mCh. There is no significant difference among the viability change associated with mCh knockdown by the 3 Cas13d systems. Collateral activity = $(\text{GFP MFI}_{\text{no gRNA}} - \text{GFP MFI}_{\text{mCh gRNA}}) / \text{GFP MFI}_{\text{no gRNA}}$. Normalized viability = $\text{Luminescence}_{\text{no gRNA}} / \text{Luminescence}_{\text{mCh gRNA}}$.

C. In U87 cells, there is no significant difference in the collateral activity and changes in viability associated with mCh-targeting knockdown by the WT, Dano-inhibited and GA-inducible split RfxCas13d. Collateral activity = $(\text{GFP MFI}_{\text{no gRNA}} - \text{GFP MFI}_{\text{mCh gRNA}}) / \text{GFP MFI}_{\text{no gRNA}}$.

Normalized viability = $(100\% - \% \text{ zombie dead cell stain positive cells}_{\text{no gRNA}}) / (100\% - \% \text{ zombie dead cell stain positive cells}_{\text{mCh gRNA}})$.

D. In U87 cells, Dano-inhibited split PspCas13b shows reduced collateral activity compared to the WT PspCas13b, while showing no significant change in the mCh knockdown associated viability change. Collateral activity = $(\text{GFP MFI}_{\text{no gRNA}} - \text{GFP MFI}_{\text{mCh gRNA}}) / \text{GFP MFI}_{\text{no gRNA}}$.

Normalized viability = $(100\% - \% \text{ zombie dead cell stain positive cells}_{\text{no gRNA}}) / (100\% - \% \text{ zombie dead cell stain positive cells}_{\text{mCh gRNA}})$.

References

1. Abudayyeh, O. O. *et al.* C2c2 is a single-component programmable RNA-guided RNA-targeting CRISPR effector. *Science* **353**, (2016).
2. Cox, D. B. T. *et al.* RNA editing with CRISPR-Cas13. *Science* **358**, 1019–1027 (2017).
3. Abudayyeh, O. O. *et al.* RNA targeting with CRISPR-Cas13. *Nature* **550**, 280–284 (2017).
4. Konermann, S. *et al.* Transcriptome Engineering with RNA-Targeting Type VI-D CRISPR Effectors. *Cell* **173**, 665-676.e14 (2018).
5. Wang, Q. *et al.* The CRISPR-Cas13a Gene-Editing System Induces Collateral Cleavage of RNA in Glioma Cells. *Adv. Sci.* **6**, 1901299 (2019).
6. Tong, H. *et al.* High-fidelity Cas13 variants for targeted RNA degradation with minimal collateral effects. *Nat Biotechnol* **41**, 108–119 (2023).
7. Shi, P. *et al.* RNA-guided cell targeting with CRISPR/RfxCas13d collateral activity in human cells. <http://biorxiv.org/lookup/doi/10.1101/2021.11.30.470032> (2021)
doi:10.1101/2021.11.30.470032.
8. Ai, Y., Liang, D. & Wilusz, J. E. CRISPR/Cas13 effectors have differing extents of off-target effects that limit their utility in eukaryotic cells. *Nucleic Acids Research* **50**, e65–e65 (2022).
9. Özcan, A. *et al.* Programmable RNA targeting with the single-protein CRISPR effector Cas7-11. *Nature* **597**, 720–725 (2021).
10. Bot, J. F., Van Der Oost, J. & Geijsen, N. The double life of CRISPR–Cas13. *Current Opinion in Biotechnology* **78**, 102789 (2022).

Reviewers' Comments:

Reviewer #1:

Remarks to the Author:

The authors have addressed my questions about the collateral activity and showed that inducible systems can help to mitigate this potential collateral activity under certain circumstances. It is well-discussed with two new supplementary figures. However, I would suggest clarifying further my second request about the possible explanation on the decrease of leakiness using NES and NLS domains. The authors speculate with two possible explanations, but I believe that an immunofluorescence and a western blot of the split and the WT systems could help to shed light to this question. Maybe just a couple of examples where each part of the protein has a different tag could help to see where and how efficiently the protein is reconstituted from the different split approaches in comparison to the WT Cas13 (in terms of amount of functional and reconstituted protein). This is especially interesting for PspCas13b that showed a lower collateral activity in the inducible approach, and it would be nice to see how this method helps to reduce this collateral effect.

Finally, there are some figures with less than 10 individual data points that are not shown.

Reviewer #2:

Remarks to the Author:

The authors have addressed the reviewers' comments by performing new experiments and rewriting parts of the manuscript. With these changes, the concerns are adequately addressed and acceptance is recommended

Reviewer #3:

Remarks to the Author:

I'd like to thank the authors for the detailed discussion and satisfyingly addressing my question. The manuscript is ready for publication in my opinion.

Response to reviewers' comments

Reviewer 1:

The authors have addressed my questions about the collateral activity and showed that inducible systems can help to mitigate this potential collateral activity under certain circumstances. It is well-discussed with two new supplementary figures. However, I would suggest clarifying further my second request about the possible explanation on the decrease of leakiness using NES and NLS domains. The authors speculate with two possible explanations, but I believe that an immunofluorescence and a western blot of the split and the WT systems could help to shed light to this question. Maybe just a couple of examples where each part of the protein has a different tag could help to see where and how efficiently the protein is reconstituted from the different split approaches in comparison to the WT Cas13 (in terms of amount of functional and reconstituted protein). This is especially interesting for PspCas13b that showed a lower collateral activity in the inducible approach, and it would be nice to see how this method helps to reduce this collateral effect.

Response:

In response to your suggestion regarding the application of staining or western blot with our system, we acknowledge this is an interesting question. To address this question, we will need to incorporate a specific epitope tag on our Cas13 fragments. However, it's important to note that the presence of epitope tags may impact the structure, stability, and function of the protein of interest¹⁻³. We anticipate a multi-step process involving redesigning inducible split Cas13 constructs with epitope tags, performance verification of new designs, and iterations incorporating different tag placements, numbers, or types for achieving efficient Cas13 activity and staining ability. The entire process, we estimate, will take weeks to months. Understanding how NES and NLS impact split protein function is not the goal of our manuscript

Conversely, the spatial separation approach has been widely utilized to minimize leakiness in split enzymes, particularly in split CRISPR/Cas9 systems. Like others in this area of study, our study focuses on improving the dynamic range of inducible enzymatic activity. And success of any optimization approach is reflected in reporter expression knockdown in our study, analogous to indels for split Cas9⁴ and gene activation/inhibition for dCas9 activator/repressors^{5,6}. Our narrative constructs a functional toolkit with exceptional performances, comprehensive characterization, and a demonstration of the robustness and applicability of such tools.

None of the studies that utilized NES/NLS tagging to minimize split enzyme leaky activity present data showing the distribution and localization of the tagged protein fragments⁴⁻⁶. However, we found data in studies focusing on characterizing dimerization systems. We've identified three relevant studies that align with our anticipated outcomes and support our hypothesis:

- a. Miyamoto et al.'s demonstration of gibberellic acid (GA) inducing colocalization of a GID domain targeted to the cytosol and an NLS-tagged GAI domain, showcasing uniform expression in the nucleus and cytosol upon GA induction⁷ (supplementary figure 10).
- b. Liang et al.'s study on ABA-induced colocalization of ABI and PYL domains, where the cytosol-targeted PYL domain moves into the nucleus upon ABA induction⁸ (Figure S11).

- c. Foight et al.'s investigation of induced and inhibited dimerization of NS3/DNCR and NS3/ANR domains supporting our hypothesis on the availability of NLS-tagged fragments in the cytosol⁹ (same dano induced and inhibited CIDs as in our study).

Finally, in response to review 1's original question on the mechanism behind the dimerization of the differentially targeted fragments, we posit that staining and western blot would not reveal the mechanism and that a mechanistic study is limited in its applicability to future designs of similar systems, as NLS and NES tagging have been tailored to specific systems. For example, our study consistently screened multiple NES/NLS architectures with all inducible split Cas13 versions. We observed that the NLS/NES architecture generating the best performance is inconsistent across different CIDs and Cas13 orthologs. Hence, a detailed study on the distribution of split fragments using staining may not unveil generalizable design rules.

Finally, there are some figures with less than 10 individual data points that are not shown.

Response:

Thank you, and we have replaced supplementary Figure 2 and 10. Please note that even though split site screening data appears to be bar charts, they are XY plots with x-axis being the amino acid position for split sites and y-axis being the relative mCherry expression.

References:

1. Booth, W. T. *et al.* Impact of an N-terminal Polyhistidine Tag on Protein Thermal Stability. *ACS Omega* **3**, 760–768 (2018).
2. Goel, A. *et al.* Relative position of the hexahistidine tag effects binding properties of a tumor-associated single-chain Fv construct. *Biochimica et Biophysica Acta (BBA) - General Subjects* **1523**, 13–20 (2000).
3. Bucher, M. H., Evdokimov, A. G. & Waugh, D. S. Differential effects of short affinity tags on the crystallization of *Pyrococcus furiosus* maltodextrin-binding protein. *Acta Crystallogr D Biol Crystallogr* **58**, 392–397 (2002).
4. Zetsche, B., Volz, S. E. & Zhang, F. A split-Cas9 architecture for inducible genome editing and transcription modulation. *Nature Biotechnology* **33**, 139–142 (2015).
5. Yu, Y. *et al.* Engineering a far-red light-activated split-Cas9 system for remote-controlled genome editing of internal organs and tumors. *Sci. Adv.* **6**, eabb1777 (2020).
6. Konermann, S. *et al.* Optical control of mammalian endogenous transcription and epigenetic states. *Nature* **500**, 472–476 (2013).
7. Miyamoto, T. *et al.* Rapid and orthogonal logic gating with a gibberellin-induced dimerization system. *Nature Chemical Biology* **8**, 465–470 (2012).
8. Liang, F. S., Ho, W. Q. & Crabtree, G. R. Engineering the ABA Plant stress pathway for regulation of induced proximity. *Science Signaling* **4**, 1–10 (2011).

9. Foight, G. W. *et al.* Multi-input chemical control of protein dimerization for programming graded cellular responses. *Nat Biotechnol* **37**, 1209–1216 (2019).